# Physically Valid Biomolecular Interaction Modeling with Gauss-Seidel Projection

**Siyuan Chen**[1,*]    **Minghao Guo**[2,*,†]    **Caoliwen Wang**[1]    **Anka He Chen**[3]
**Yikun Zhang**[4]    **Jingjing Chai**[5]    **Yin Yang**[6]    **Wojciech Matusik**[2]    **Peter Yichen Chen**[1,†]
[1]University of British Columbia    [2]MIT CSAIL    [3]NVIDIA    [4]Peking University
[5]Foundry Biosciences    [6]University of Utah

## Abstract

Biomolecular interaction modeling has been substantially advanced by foundation models, yet they often produce all-atom structures that violate basic steric feasibility. We address this limitation by enforcing physical validity as a strict constraint during both training and inference with a unified module. At its core is a differentiable projection that maps the provisional atom coordinates from the diffusion model to the nearest physically valid configuration. This projection is achieved using a Gauss-Seidel scheme, which exploits the locality and sparsity of the constraints to ensure stable and fast convergence at scale. By implicit differentiation to obtain gradients, our module integrates seamlessly into existing frameworks for end-to-end finetuning. With our Gauss-Seidel projection module in place, two denoising steps are sufficient to produce biomolecular complexes that are both physically valid and structurally accurate. Across six benchmarks, our 2-step model achieves the same structural accuracy as state-of-the-art 200-step diffusion baselines, delivering $\sim 10\times$ wall-clock speedups while guaranteeing physical validity. The code is available at `https://github.com/chensiyuan030105/ProteinGS.git`.

## 1 Introduction

End-to-end, all-atom protein structure predictors that integrate deep learning with generative modeling are emerging as transformative tools for biomolecular interaction modeling (Senior et al., 2020b; Jumper et al., 2021; Abramson et al., 2024; Corso et al., 2023; Watson et al., 2023). These systems achieve unprecedented accuracy in predicting arbitrary biomolecular complexes and are shaping the future of computational biology and drug discovery.

Contrary to their high structural accuracy, current predictors often fail to satisfy a conceptually simpler but basic requirement: the *physical validity* of the all-atom output. This failure, also commonly termed hallucination, appears as steric clashes, distorted covalent geometry, and stereochemical errors (Fig. 11). In practice, however, physical validity is a prerequisite: Atom-level physics violations hinder expert assessment (Senior et al., 2020a), undermine structure-based reasoning and experimental planning (Lyu et al., 2019), and also destabilize downstream computational analyses such as molecular dynamics (Hollingsworth & Dror, 2018; Lindorff-Larsen et al., 2011).

The root cause of this problem lies in the design of current predictors: as generative models, they are trained to match the empirical distribution of known structures, without enforcing physical validity as a strict constraint in the training objective. Consequently, these models can assign non-zero probability to non-physical configurations, a flaw that persists even with large-scale training, both in data and model size, as observed in recent works (Wohlwend et al., 2024; Passaro et al., 2025; Buttenschoen et al., 2024; Team et al., 2025). Methods such as Boltz-1-Steering (Wohlwend et al., 2024) reintroduce physics as inference-time guidance by biasing the sampling process towards physically valid regions. Such guidance can reduce violations, but still cannot guarantee validity: with finite guidance strength and update steps, invalid configurations remain reachable.

---

[*]Equal contribution.
[†]Corresponding author.

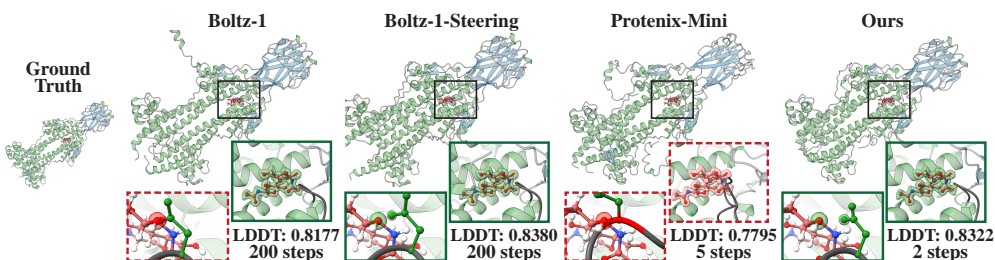

Figure 1: **Comparison on PDB 8B3E** (protein-ligand complex): global structure (top) and two zoomed views of the binding pocket (bottom). Protenix-Mini with 5 denoising steps exhibits backbone-ligand clashing. With 200 steps, Boltz-1 resolves the backbone but leaves clashes between the side chain and the ligand. Boltz-1-Steering removes clashes by using physics-informed potentials, but at the cost of large sampling steps. Ours yields physically valid results with only 2 denoising steps.

To close this gap, we elevate physical validity to a *first-class constraint* and enforce it in *both training and inference* by explicitly handling validity alongside generation: The denoising network first outputs provisional atom coordinates; a separate projection module then maps them onto the physically valid set. Losses are computed on the projected coordinates with gradients propagated through the projection module. This design allows the denoising network to redirect its capacity toward improving structural accuracy, offloading the task of avoiding physics violations to the projection module. This shift further removes the need for large denoising steps: at inference, sampling with few steps is sufficient to attain high structural accuracy with guaranteed physical validity. As such, the reduction in denoising steps is not due to specialized few-step diffusion techniques: it follows directly from treating physical validity as a constraint throughout both training and inference.

Concretely, we instantiate this decoupling with a *differentiable Gauss-Seidel projection* placed after the denoiser. It solves a constrained optimization that projects all-atom coordinates to the nearest physically valid configuration. The constraints are tightly coupled through shared atoms and their number scales with the number of atoms, making the optimization problem large. Within the training loop, the module must also converge in a handful of iterations with low memory per forward and backward pass. These requirements make pure first-order gradient descent methods impractical, which require tiny steps and many iterations to converge. We therefore adopt a Gauss-Seidel scheme (Saad, 2003), which sweeps over all constraints for a few iterations, enforcing each locally by updating only the affected atoms. It leverages locality and sparsity of the constraints and thereby yields faster, more stable convergence than gradient descent. The projection module is differentiable via implicit differentiation. We integrate it as a drop-in layer within existing biomolecular interaction frameworks (e.g., Boltz (Wohlwend et al., 2024)) and finetune end-to-end. At inference time, 2 denoising steps suffice to produce structurally accurate, physically valid protein complexes.

We evaluate our method on six protein-complex benchmarks: CASP15, Test (Wohlwend et al., 2024), PoseBusters (Buttenschoen et al., 2024), AF3-AB (Abramson et al., 2024), dsDNA, and RNA-Protein (Ma et al., 2025). Comparisons include 200-step generative models (Boltz-1, Boltz-1-Steering (Wohlwend et al., 2024), Boltz-2 (Passaro et al., 2025), and Protenix (Team et al., 2025)) and the few-step baseline Protenix-Mini (Gong et al., 2025). Despite using only two denoising steps, our model achieves competitive structural accuracy while guaranteeing physical validity. For runtime, our model delivers a $\sim 10\times$ wall-clock speedup over baselines. Our study largely closes the gap between guaranteed physical validity and state-of-the-art structural accuracy under few-step sampling, enabling 2-step all-atom predictions with an order-of-magnitude faster inference.

## 2 RELATED WORK

**Protein Structure Prediction.** Deep learning-based foundation models are reshaping the task of protein structure prediction that produces 3D coordinates from input sequences (Senior et al., 2020b; Jumper et al., 2021). End-to-end variants extended this paradigm to complexes and large-scale training and inference (Baek et al., 2021; Evans et al., 2022). AlphaFold-3 further moved to all-atom biomolecular interaction modeling with a diffusion sampler (Abramson et al., 2024), followed by improvements on generative models, such as Boltz-2 (Passaro et al., 2025) and Protenix (Team et al., 2025). These methods typically require hundreds of denoising steps. Few-step variants such

as Protenix-Mini (Gong et al., 2025) reduce sampling to two steps while maintaining accuracy, yet still lack guarantees on physical validity. Boltz-1-Steering (Wohlwend et al., 2024) is the first to explicitly target this issue by steering sampling with physics-informed potentials, but the guidance is soft and cannot preclude violations. By contrast, we introduce a Gauss-Seidel projection layer that enforces physical validity for both training and inference.

**Diffusion Models with Tilted Distribution Sampling.** Diffusion models have demonstrated strong capabilities in producing samples given input conditions (Ho et al., 2020; Dhariwal & Nichol, 2021; Song et al., 2021). However, in many cases, users require the generated samples to satisfy certain specific constraints. There are primarily two types of constraint-aware diffusion methods: (i) Feynman-Kac (FK) steering-based conditional sampling methods (Trippe et al., 2023; Singhal et al., 2025), which steer the sampling toward regions that meet desired conditions through reweighting of the path distribution in the diffusion process, and (ii) manifold-based diffusion models (Bortoli et al., 2022; Elhag et al., 2024), which explicitly construct a constrained manifold, ensuring that generated samples inherently satisfy the desired constraints. However, FK-steering typically requires a large number of sampling steps, and explicit manifold construction can be computationally expensive when constraints are complex. We introduce a Gauss-Seidel projection module that enforces constraints in training and inference, thereby circumventing both excessive sampling iterations and the need for expensive manifold construction.

**Constraint Enforcement via Position-Based Dynamics.** Enforcing physical constraints has long been central to physics simulation across computational physics and computer graphics. To handle boundary conditions (Macklin et al., 2014), collisions, contact, friction (Bridson et al., 2002), and bond stretching (Ryckaert et al., 1977), many methods have been developed to keep simulations physically consistent. Among these, position-based dynamics and its extensions (Müller et al., 2007; Macklin & Müller, 2016; Chen et al., 2024) iteratively project particle positions to satisfy predefined constraints, proving effective for soft bodies, fluids, and coupled phenomena (Bender et al., 2014). By updating positions directly instead of integrating stiff forces, these methods remain stable under large timesteps and avoid costly global solves. Since constraints are enforced through small, local projections, the overall complexity scales nearly linearly with the number of constraints, which aligns with our goal of generating proteins that respect local physical constraints.

**Finetuning Diffusion Models.** Finetuning has become essential for tailoring diffusion models to specific downstream tasks. To enhance controllability, many recent works employ reinforcement learning (RL) strategies: (i) methods such as DRaFT (Clark et al., 2024) and DPPO (Ren et al., 2024) treat the denoising trajectory as a policy to directly optimize reward functions; (ii) approaches like DPOK (Fan et al., 2023) and DiffPPO (Xiao et al., 2024) leverage preference data or human feedback. While effective at steering generation, these RL-based methods often incur high computational costs or exhibit sensitivity to the quality of the feedback signal. In contrast, we introduce a Gauss-Seidel projection during training to finetune the model via strictly defined physical constraints, circumventing the instability of policy gradient estimation.

## 3 PRELIMINARY

**All-Atom Diffusion Model.** Proteins can be represented at *all-atom* resolution (Abramson et al., 2024; Wohlwend et al., 2024; Passaro et al., 2025), where a structure is specified by the Cartesian coordinates of all atoms $\hat{\mathbf{x}} \in \mathbb{R}^{N \times 3}$, with $N$ the total number of atoms. This representation enables direct modeling of side-chain orientations, local packing, and explicit inter-atomic geometry. AlphaFold3 (Abramson et al., 2024) introduced an atom-level diffusion approach that iteratively denoises atomic coordinates from Gaussian noise. Conditioned on features produced by the atom-attention encoder, MSA module, and PairFormer, a denoising network runs for hundreds of steps (typically 200) to generate the all-atom prediction $\hat{\mathbf{x}}$.

**Physics-Guided Steering.** All-atom diffusion sampling is stochastic and therefore cannot guarantee *physical validity*. To address this, Boltz-1-Steering (Wohlwend et al., 2024), built upon the Feynman-Kac framework (Singhal et al., 2025), employs physics-informed potentials. At each denoising step, these potentials score the validity and tilt the learned denoising model toward a lower-energy configuration. The energy is a weighted sum of seven terms: tetrahedral atom chirality, bond stereochemistry, planarity of double bonds, internal ligand distance bounds, steric clashes avoidance, non-overlapping symmetric chains, and the preservation of covalently bonded chains.

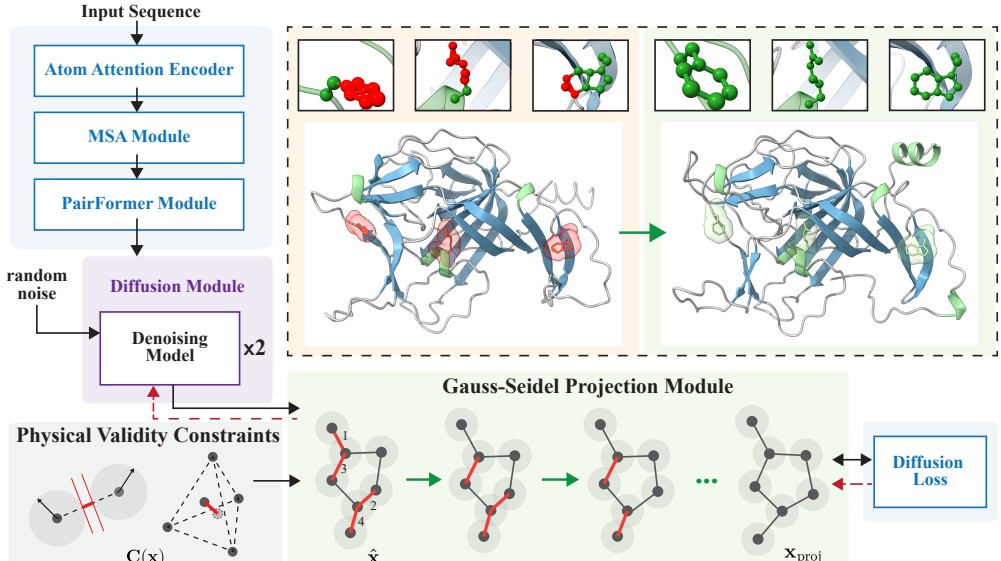

Figure 2: **Enforcing physical validity during both training and inference via our Gauss-Seidel projection module.** Provisional all-atom coordinates from the diffusion model are corrected by a Gauss-Seidel projection that sequentially resolves local constraints, each acting on a small set of atoms and updating coordinates in place. The module is differentiable via implicit differentiation, allowing seamless integration into training. The same projection is applied at inference, ensuring physical validity and enabling accurate predictions with as few as two denoising steps.

Since this tilted distribution cannot be sampled directly, the method uses importance sampling to draw multiple candidates, which are then refined with gradient descent updates to further reduce violations. Although this steering reduces physical errors, it remains a soft approach: it biases the sampling process toward validity during inference rather than embedding validity into the training.

## 4 APPROACH

Our goal is to predict all-atom biomolecular complexes that are both structurally accurate and physically valid. Given provisional coordinates produced by a diffusion module, we enforce validity with a Gauss-Seidel projection module that sequentially resolves physical constraints on the affected atoms (Sec. 4.1 and 4.2). The module is differentiable, allowing training losses to be computed on the projected atom coordinates, and gradients to propagate back through the projection module via implicit differentiation (Sec. 4.3). Crucially, this module enables accurate inference with only two denoising steps while guaranteeing physical validity. Fig. 2 illustrates the overall pipeline.

### 4.1 PENALTY-BASED FORMULATION FOR PHYSICAL VALIDITY

Given the Cartesian coordinates of all constituent atoms $\hat{\mathbf{x}}$ produced by the diffusion module, our projection module finds the nearest physically valid coordinate configuration:

$$\mathbf{x}_{\text{proj}} = \arg\min_{\mathbf{x}} \tfrac{1}{2}\|\mathbf{x} - \hat{\mathbf{x}}\|_2^2, \qquad \text{s.t. } \mathbf{C}(\mathbf{x}) = \mathbf{0}, \tag{1}$$

where $\mathbf{C}(\cdot) : \mathbb{R}^{N\times 3} \to \mathbb{R}^m$ concatenates all constraints that characterize physical validity, and its zero set defines the feasible space. We follow Boltz-1-Steering (Wohlwend et al., 2024) to define the constraints. Detailed formulation is provided in Appendix L. Since all terms are locally defined on pairs or four-atom groups, the total number of constraints $m$ scales with atoms and can be large. For instance, for PDB 8TGH which consists of $470$ residues with $7,082$ atoms, $m$ is $\sim$13M.

A standard approach to solving the constrained problem Eq. 1 is the *penalty method* (Boyd & Vandenberghe, 2004), in which the constraints are incorporated as an exterior penalty into the objective:

$$\mathbf{x}_{\text{proj}} = \arg\min_{\mathbf{x}}\big(E(\mathbf{x}) + \tfrac{1}{2}\|\mathbf{x} - \hat{\mathbf{x}}\|_2^2\big), \qquad E(\mathbf{x}) = \tfrac{1}{2}\mathbf{C}(\mathbf{x})^\top \boldsymbol{\alpha}^{-1}\mathbf{C}(\mathbf{x}), \tag{2}$$

where we choose a quadratic penalty $E(\cdot)$ and $\boldsymbol{\alpha} \in \mathbb{R}^{m \times m}$ is a block-diagonal matrix of penalty coefficients. For sufficiently small $\boldsymbol{\alpha}$ (i.e., large weights $\boldsymbol{\alpha}^{-1}$), Eq. 2 serves as a numerical realization of the hard-constrained projection: the minimizer satisfies the constraint up to numerical tolerance. In our implementation, we set $\boldsymbol{\alpha}_j = 10^{-7}$ or $10^{-6}$ to enforce a tight feasibility threshold.

The penalty formulation defines a deterministic mapping $\hat{\mathbf{x}} \mapsto \mathbf{x}_{\mathrm{proj}}$ that links the provisional coordinates of the diffusion module to a physically valid output. Solved on its own, this mapping already functions as a plug-and-play post-processing module for biomolecular modeling frameworks. Since the projection can resolve invalidity efficiently, we aim to explicitly decouple validity enforcement from the network training by integrating the module *within* the training loop. Concretely, we insert the module immediately after the diffusion module and compute training losses on the projected coordinates. In the following sections, we detail the forward solver (Sec. 4.2) and the backward pass via implicit differentiation (Sec. 4.3), enabling the module to operate as a fully differentiable layer.

## 4.2 GAUSS-SEIDEL CONSTRAINT PROJECTION

Solving Eq. 2 alone is classical and admits many off-the-shelf solvers. Embedding it as a differentiable module within a biomolecular modeling pipeline, however, imposes stricter requirements: very fast convergence per forward pass and training step, and stable convergence under stiff, tightly coupled constraints. Pure first-order gradient descent, as used in Boltz-1-Steering (Wohlwend et al., 2024), is ill-suited: it requires small step sizes and prohibitively many iterations to converge to a zero constraint penalty, making unrolled end-to-end training slow. We therefore propose a Gauss-Seidel projection that exploits locality and sparsity by sequentially updating only the atoms affected by each constraint, thereby achieving fast and stable convergence.

We consider the first-order optimality condition for Eq. 2:

$$\nabla \mathbf{C}(\mathbf{x})^{\top} \boldsymbol{\alpha}^{-1} \mathbf{C}(\mathbf{x}) + (\mathbf{x} - \hat{\mathbf{x}}) = \mathbf{0}. \tag{3}$$

We then introduce the Lagrange multiplier $\boldsymbol{\lambda}(\mathbf{x}) := -\boldsymbol{\alpha}^{-1} \mathbf{C}(\mathbf{x})$ by following (Stuart & Humphries, 1996) and (Servin et al., 2006) to obtain the coupled system:

$$\begin{aligned} (\mathbf{x} - \hat{\mathbf{x}}) - \nabla \mathbf{C}(\mathbf{x})^{\top} \boldsymbol{\lambda}(\mathbf{x}) &= \mathbf{0}, \\ \mathbf{C}(\mathbf{x}) + \boldsymbol{\alpha}\, \boldsymbol{\lambda}(\mathbf{x}) &= \mathbf{0}. \end{aligned} \tag{4}$$

This system is nonlinear; we solve it by iterative linearization about the current iterate $(\mathbf{x}^{(n)}, \boldsymbol{\lambda}^{(n)})$. At iteration $n$, the linearized system is:

$$\begin{bmatrix} \mathbf{I} & -\nabla \mathbf{C}(\mathbf{x}^{(n)})^{\top} \\ \nabla \mathbf{C}(\mathbf{x}^{(n)}) & \boldsymbol{\alpha} \end{bmatrix} \begin{bmatrix} \Delta \mathbf{x} \\ \Delta \boldsymbol{\lambda} \end{bmatrix} = - \begin{bmatrix} \mathbf{0} \\ \mathbf{C}(\mathbf{x}^{(n)}) + \boldsymbol{\alpha}\, \boldsymbol{\lambda}^{(n)} \end{bmatrix},$$

with updates $\mathbf{x}^{(n+1)} = \mathbf{x}^{(n)} + \Delta \mathbf{x}$ and $\boldsymbol{\lambda}^{(n+1)} = \boldsymbol{\lambda}^{(n)} + \Delta \boldsymbol{\lambda}$. It can be shown that with initialization $\mathbf{x}^{(0)} = \hat{\mathbf{x}}$ and $\boldsymbol{\lambda}^{(0)} = \mathbf{0}$, the linearized iterates converge to a solution of the original nonlinear system Eq. 4. See Appendix I for a proof. Applying the Schur complement (Zhang, 2005) with respect to $\mathbf{I}$ gives the reduced system for the multiplier update:

$$\left(\nabla \mathbf{C}(\mathbf{x}^{(n)}) \nabla \mathbf{C}(\mathbf{x}^{(n)})^{\top} + \boldsymbol{\alpha}\right) \Delta \boldsymbol{\lambda} = -\mathbf{C}(\mathbf{x}^{(n)}) - \boldsymbol{\alpha}\, \boldsymbol{\lambda}^{(n)}, \tag{5}$$

and the position update $\Delta \mathbf{x} = \nabla \mathbf{C}(\mathbf{x}^{(n)})^{\top} \Delta \boldsymbol{\lambda}$.

The primary computation in each iteration is solving the large linear system Eq. 5. Rather than constructing the full linear system, which is memory-intensive and costly, we employ a *Gauss-Seidel* scheme that iteratively sweeps over the constraints: During each sweep, the method addresses one constraint at a time by updating only the atom coordinates it affects. These new coordinates are used immediately when processing the next constraint. After a small number of sweeps, the system converges to a configuration where all constraints are satisfied. This Gauss-Seidel scheme is significantly more efficient than a global solve because it effectively leverages the locality and sparsity inherent in Eq. 5. Specifically, for the $j$-th constraint, the Gauss-Seidel update is

$$\Delta \boldsymbol{\lambda}_j = \frac{-\mathbf{C}_j(\mathbf{x}^{(n)}) - \boldsymbol{\alpha}_j \boldsymbol{\lambda}_j^{(n)}}{\nabla \mathbf{C}_j(\mathbf{x}^{(n)}) \nabla \mathbf{C}_j(\mathbf{x}^{(n)})^{\top} + \boldsymbol{\alpha}_j}, \qquad j = 1, ..., m. \tag{6}$$

**Algorithm 1** Forward Solver of Gauss-Seidel Projection Module.

---

**Input:** Atom coordinates $\hat{\mathbf{x}}$; constraints and their gradients $\{\mathbf{C}_j, \nabla\mathbf{C}_j\}_{j=1}^m$; penalty coefficients $\{\alpha_j\}_{j=1}^m$; sweeps $T$
**Output:** Projected coordinates $\mathbf{x}_{\text{proj}}$
1: $\mathbf{x}^{(0)} \leftarrow \hat{\mathbf{x}}; \quad \boldsymbol{\lambda}^{(0)} \leftarrow \mathbf{0}$
2: **for** $n = 1$ to $T$ **do**     $\triangleright$ Gauss–Seidel sweeps
3:     **for** $j = 1$ to $m$ **do**
4:         Compute $\Delta\boldsymbol{\lambda}_j$ using Eq. 6.
5:         $\mathbf{x}^{(n+1)} \leftarrow \mathbf{x}^{(n)} + \nabla\mathbf{C}_j(\mathbf{x}^{(n)})\Delta\boldsymbol{\lambda}_j$;
6:         $\boldsymbol{\lambda}_j^{(n+1)} \leftarrow \boldsymbol{\lambda}_j^{(n)} + \Delta\boldsymbol{\lambda}_j$
7:     **end for**
8: **end for**
9: **return** $\mathbf{x}_{\text{proj}} \leftarrow \mathbf{x}^{(T)}$

**Algorithm 2** Backward Pass of Gauss-Seidel Projection Module.

---

**Input:** Projected coords $\mathbf{x}_{\text{proj}}$; upstream gradient $\partial L/\partial\mathbf{x}_{\text{proj}}$; $\{\mathbf{C}_j, \nabla\mathbf{C}_j, \nabla^2\mathbf{C}_j\}_{j=1}^m$; $\{\alpha_j\}_{j=1}^m$; tolerance $\varepsilon$; max iters $K$
**Output:** Gradient w.r.t. $\hat{\mathbf{x}}$, i.e., $\partial L/\partial\hat{\mathbf{x}}$
1: $\mathbf{z}_0 \leftarrow (\partial L/\partial\mathbf{x}_{\text{proj}})^\top$
2: Compute $\mathbf{A} = \mathbf{H}(\mathbf{x}_{\text{proj}}) + \mathbf{I}$ as in Eq. 7
3: **for** $k = 1$ to $K$ **do**
4:     CG update step for $\mathbf{A}^\top\mathbf{z}_k = \left(\frac{\partial L}{\partial\mathbf{x}_{\text{proj}}}\right)^\top$
5:     **if** $\|\mathbf{z}_k - \mathbf{z}_{k-1}\| \leq \varepsilon$ **then**
6:         **break**
7:     **end if**
8: **end for**
9: **return** $\frac{\partial L}{\partial\hat{\mathbf{x}}} \leftarrow \mathbf{z}^\top$

Processing all constraints sequentially and repeating for a small number of sweeps (20 in our implementation) yields fast and stable convergence in practice. To maximize computational throughput, we implement the solver on GPUs (details in Sec. 5.1). Algorithm 1 outlines the forward process of the Gauss-Seidel projection module.

**Remark 4.1 (Fast Convergence and Constraint Satisfaction of Gauss-Seidel Projection)** *The convergence of the Gauss-Seidel projection is of quasi-second order. Each update yields a monotone decrease with $O(1)$ work per constraint (hence $O(m)$ per sweep). With a strict tolerance and sufficiently small $\boldsymbol{\alpha}$, the final iterate satisfies the first-order optimality condition and realizes the hard-constrained projection. See Appendix J for details.*

## 4.3 DERIVATIVES VIA IMPLICIT DIFFERENTIATION

In the forward pass, the module projects $\hat{\mathbf{x}}$ into $\mathbf{x}_{\text{proj}} = \mathbf{x}_{\text{proj}}(\hat{\mathbf{x}})$, which is used to calculate the training loss $L = L(\mathbf{x}_{\text{proj}})$. The corresponding backward pass is to compute the gradient of this loss with respect to the input of the projection module, $\frac{\partial L}{\partial\hat{\mathbf{x}}}$.

Making the projection module differentiable is the key to enabling few-step sampling. A standard diffusion module is trained to handle two tasks simultaneously: achieving structural accuracy and ensuring physical validity. It relies on a large number of denoising steps as its effective capacity to perform both. Consequently, when the step budget is small, the model's structural accuracy degrades significantly, as shown in Fig. 3. By finetuning the network with our differentiable projection module, we enable a decoupling of responsibilities. The denoising network learns to focus exclusively on recovering structural accuracy, offloading the task of ensuring physical validity to the projection module. This decoupling makes highly accurate predictions possible even with a very small step budget.

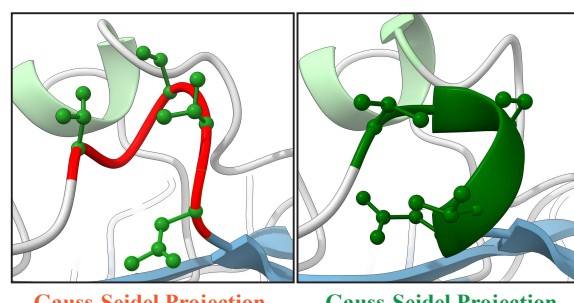

**Gauss-Seidel Projection w/o Finetuning**          **Gauss-Seidel Projection w/ Finetuning**

Figure 3: **Importance of differentiable projection and finetuning.** A post-hoc projection without finetuning with a 2-step sampling, while ensuring physical validity, fails to recover the $\alpha$-helical secondary structure (left). Integrating the projection as a differentiable layer and finetuning diffusion module restores the helix and improves structural accuracy (right).

Backpropagating through the Gauss-Seidel projection is non-trivial because it is implemented with an iterative solver. A naive approach, such as unrolling the forward iterations for automatic differen-

tiation, is prohibitively memory-intensive and thus impractical for end-to-end training. We resort to *implicit differentiation*, a technique from sensitivity analysis (Burczyński et al., 1997). Specifically, we differentiate the first-order optimality condition in Eq. 2 with respect to $\hat{\mathbf{x}}$:

$$\left(\mathbf{H}(\mathbf{x}_{\text{proj}}) + \mathbf{I}\right) \frac{\partial \mathbf{x}_{\text{proj}}}{\partial \hat{\mathbf{x}}} = \mathbf{I}, \quad \mathbf{H}(\mathbf{x}) = \sum_{j=1}^{m} \alpha_j^{-1} \left[ \nabla \mathbf{C}_j(\mathbf{x}) \nabla \mathbf{C}_j(\mathbf{x})^\top + \mathbf{C}_j(\mathbf{x}) \nabla^2 \mathbf{C}_j(\mathbf{x}) \right], \quad (7)$$

which is evaluated at $\mathbf{x} = \mathbf{x}_{\text{proj}}$. By the chain rule $\frac{\partial L}{\partial \hat{\mathbf{x}}} = \frac{\partial L}{\partial \mathbf{x}_{\text{proj}}} \frac{\partial \mathbf{x}_{\text{proj}}}{\partial \hat{\mathbf{x}}}$, we obtain the adjoint system

$$\left(\mathbf{H}(\mathbf{x}_{\text{proj}}) + \mathbf{I}\right)^\top \mathbf{z} = \left(\frac{\partial L}{\partial \mathbf{x}_{\text{proj}}}\right)^\top, \quad \text{and set} \quad \frac{\partial L}{\partial \hat{\mathbf{x}}} = \mathbf{z}^\top. \quad (8)$$

Analogously to the forward pass, the backward pass requires solving an additional linear system. We use conjugate gradients (CG) to solve Eq. 8, whose convergence is guaranteed near feasibility (see Appendix K for details). For runtime efficiency, we implement the CG solver on the GPU. Algorithm 2 outlines the backward process of the projection module.

## 5 EVALUATION

In this section, we evaluate our method's structural accuracy and physical validity through a series of experiments. We report quantitative results on six protein-complex benchmarks using standard evaluation metrics (Sec. 5.2). We also provide qualitative visualizations that compare protein structures and demonstrate the physical validity (Sec. 5.3). Furthermore, we analyze the convergence speed and wall-clock runtime of our approach (Sec. 5.4). Finally, we conduct an ablation study to isolate the impact of our Gauss-Seidel projection (Sec. 5.5).

### 5.1 IMPLEMENTATION DETAILS

With the differentiable Gauss-Seidel projection module in place, we finetune a pre-trained all-atom diffusion network specifically for 2-step sampling. During finetuning, we retain the original noise parameterization and denoising objective. At inference time, we use the inference-time sampling schedule of Protenix-Mini (Team et al., 2025). A Gauss-Seidel projection is applied to the model's output to guarantee a physically valid final structure. Appendix H provides pseudo-code and complete algorithms for both training and inference with our projection module. We implement the differentiable Gauss-Seidel projection on the GPU using the WARP library (Macklin et al., 2024). In the forward pass, constraints are partitioned into batches with no shared atoms, allowing them to be processed in parallel. Coordinate updates are accumulated with atomic operations. We use $T=20$ sweeps. The backward pass solves the adjoint linear system with a conjugate gradient solver fully on GPU, parallelizing per-constraint operations with atomic accumulation. We set the tolerance to $\varepsilon = 10^{-4}$ and the maximal number of iterations to 1000. The projection module integrates with PyTorch as a custom autograd layer, avoiding the overhead costs of host-device copies and context switching. For training, we finetune Boltz-1 (Wohlwend et al., 2024) from open-source pretrained weights on the RCSB dataset (Burley et al., 2022) ($\sim 180{,}000$ structures) using 8 NVIDIA H100 GPUs over two weeks. All evaluations are conducted on NVIDIA A6000 GPUs. For baselines and evaluation protocol, see Appendix M for details.

### 5.2 QUANTITATIVE RESULTS

Table 1 and 2 summarize the comparison across six protein-complex benchmarks. Among all methods, ours uses the fewest denoising steps (2 steps). It achieves state-of-the-art accuracy in the few-step regime: Compared to Protenix-Mini (5 steps), our model improves complex- and interface-level LDDT across most benchmarks. Compared to 200-step baselines, the accuracy of our method is competitive. On Test and PoseBusters, our performance is comparable to Boltz-1 and Protenix across multiple metrics while approaching Boltz-2, which requires 100 times more sampling.

For physical validity, our approach is unequivocally best: it achieves $100\%$ validity across all benchmarks. Methods that do not enforce validity (Boltz-1, Protenix, Protenix-Mini) show frequent

Table 1: **Quantitative results on 6 datasets (Part I).** Our method uses only 2 denoising steps yet always guarantees physical validity and achieves competitive structural accuracy compared to baselines. Dark green cells denote the best results among the few-step methods, while light green cells denote the best results among the 200-step methods. We also include the results of AlphaFold3 (Abramson et al., 2024) for references.

| | Method / Metric | # Denoise Steps | Complex LDDT | Prot-Prot iLDDT | Lig-Prot iLDDT | DockQ > 0.23 | L-RMSD < 2 Å | TM-score | Physical Validity |
|---|---|---|---|---|---|---|---|---|---|
| CASP15 | AlphaFold3 | 200 | 0.65±0.06 | 0.83±0.05 | 0.58±0.39 | 0.67±0.33 | 0.36±0.28 | 0.39±0.08 | 0.74±0.11 |
| | Boltz-1 | 200 | 0.64±0.06 | 0.53±0.16 | 0.58±0.37 | 0.73±0.27 | 0.20±0.20 | 0.37±0.07 | 0.65±0.13 |
| | Protenix | 200 | 0.65±0.06 | 0.75±0.17 | 0.59±0.38 | 0.56±0.33 | 0.28±0.25 | 0.38±0.08 | 0.65±0.13 |
| | Boltz-1-Steering | 200 | 0.64±0.06 | 0.45±0.11 | 0.58±0.37 | 0.71±0.29 | 0.15±0.15 | 0.37±0.07 | 1.00±0.00 |
| | Boltz-2 | 200 | 0.67±0.06 | 0.74±0.12 | 0.71±0.29 | 0.73±0.27 | 0.42±0.27 | 0.40±0.08 | 1.00±0.00 |
| | Protenix-Mini | 10 | 0.60±0.06 | 0.54±0.12 | 0.51±0.37 | 0.38±0.26 | 0.37±0.27 | 0.35±0.07 | 0.66±0.13 |
| | Protenix-Mini | 5 | 0.59±0.06 | 0.56±0.14 | 0.49±0.37 | 0.41±0.24 | 0.37±0.24 | 0.36±0.07 | 0.64±0.13 |
| | Ours | 2 | 0.63±0.06 | 0.41±0.15 | 0.52±0.36 | 0.71±0.29 | 0.36±0.28 | 0.36±0.07 | 1.00±0.00 |
| Test | AlphaFold3 | 200 | 0.82±0.01 | 0.50±0.03 | 0.58±0.07 | 0.72±0.05 | 0.57±0.06 | 0.85±0.02 | 0.62±0.02 |
| | Boltz-1 | 200 | 0.80±0.01 | 0.44±0.03 | 0.52±0.07 | 0.69±0.05 | 0.56±0.06 | 0.83±0.02 | 0.44±0.04 |
| | Protenix | 200 | 0.80±0.01 | 0.44±0.03 | 0.56±0.07 | 0.68±0.06 | 0.55±0.06 | 0.83±0.02 | 0.51±0.04 |
| | Boltz-1-Steering | 200 | 0.80±0.01 | 0.44±0.03 | 0.55±0.07 | 0.68±0.06 | 0.55±0.06 | 0.83±0.02 | 1.00±0.00 |
| | Boltz-2 | 200 | 0.83±0.01 | 0.52±0.03 | 0.62±0.08 | 0.74±0.05 | 0.60±0.06 | 0.86±0.02 | 1.00±0.00 |
| | Protenix-Mini | 10 | 0.73±0.02 | 0.37±0.03 | 0.53±0.07 | 0.55±0.06 | 0.51±0.05 | 0.76±0.02 | 0.45±0.04 |
| | Protenix-Mini | 5 | 0.73±0.02 | 0.37±0.03 | 0.50±0.07 | 0.55±0.05 | 0.47±0.06 | 0.76±0.02 | 0.45±0.04 |
| | Ours | 2 | 0.79±0.01 | 0.40±0.03 | 0.52±0.07 | 0.65±0.06 | 0.52±0.06 | 0.81±0.02 | 1.00±0.00 |
| PoseBusters | AlphaFold3 | 200 | 0.93±0.01 | 0.57±0.06 | 0.78±0.02 | 0.65±0.08 | 0.73±0.04 | 0.90±0.02 | 0.48±0.05 |
| | Boltz-1 | 200 | 0.92±0.01 | 0.50±0.06 | 0.69±0.02 | 0.59±0.09 | 0.62±0.05 | 0.88±0.02 | 0.28±0.05 |
| | Protenix | 200 | 0.92±0.01 | 0.53±0.06 | 0.73±0.02 | 0.61±0.09 | 0.67±0.04 | 0.89±0.02 | 0.49±0.05 |
| | Boltz-1-Steering | 200 | 0.92±0.01 | 0.52±0.06 | 0.69±0.02 | 0.59±0.09 | 0.62±0.05 | 0.88±0.04 | 0.95±0.03 |
| | Boltz-2 | 200 | 0.94±0.01 | 0.57±0.06 | 0.81±0.02 | 0.63±0.09 | 0.79±0.04 | 0.90±0.02 | 0.88±0.04 |
| | Protenix-Mini | 10 | 0.90±0.01 | 0.49±0.06 | 0.72±0.02 | 0.57±0.09 | 0.67±0.04 | 0.87±0.02 | 0.37±0.05 |
| | Protenix-Mini | 5 | 0.90±0.01 | 0.49±0.06 | 0.70±0.02 | 0.58±0.09 | 0.63±0.04 | 0.87±0.03 | 0.27±0.04 |
| | Ours | 2 | 0.91±0.01 | 0.50±0.06 | 0.69±0.02 | 0.59±0.09 | 0.61±0.05 | 0.88±0.02 | 1.00±0.00 |

clashes and physical errors, whereas steering-based methods (Boltz-1-Steering, Boltz-2) improve but still admit failures because validity is used as guidance rather than a constraint. These non-physical outcomes limit downstream usability, whereas our predictions satisfy validity and preserve high structural accuracy with only two denoising steps.

## 5.3 Qualitative Results

Fig. 4 and 9 (in Appendix A) present qualitative comparisons, showing that our approach consistently yields physically valid structures while preserving high accuracy. For PDB 7Y9A (with an N-acetyl-D-glucosamine ligand), which is used to assess ligand–protein interface quality, all baselines produce severe ligand–protein atom clashes, whereas our method eliminates these violations. For PDB 7XYO (557 residues with extensive secondary structure), which is used to evaluate large, structured assemblies, Boltz-1, Boltz-1-Steering, and Protenix show inter-chain collisions. Protenix-Mini avoids clashes but introduces large distortion, particularly in the second coiled segment. In contrast, our method produces clash-free structures while maintaining structural accuracy. Additional examples are provided in Appendix A.

## 5.4 Analysis

**Analysis on Convergence Speed.** We compare the convergence speed of our Gauss-Seidel projection against gradient descent guidance (Boltz-1-Steering). We add Gaussian noise to the ground truth coordinates of PDB 8X51 with four levels ($\sigma \in \{160, 120, 80, 40\}$) and run the same diffusion module, with either gradient descent guidance or Gauss-Seidel projection to enforce physical validity. We visualize the potential energy versus the number of update iterations in Fig. 5 (left). Gauss-Seidel projection reduces the physics potential rapidly and converges within 20 iterations for all noise levels. In contrast, gradient descent exhibits oscillations and long tails, requiring far

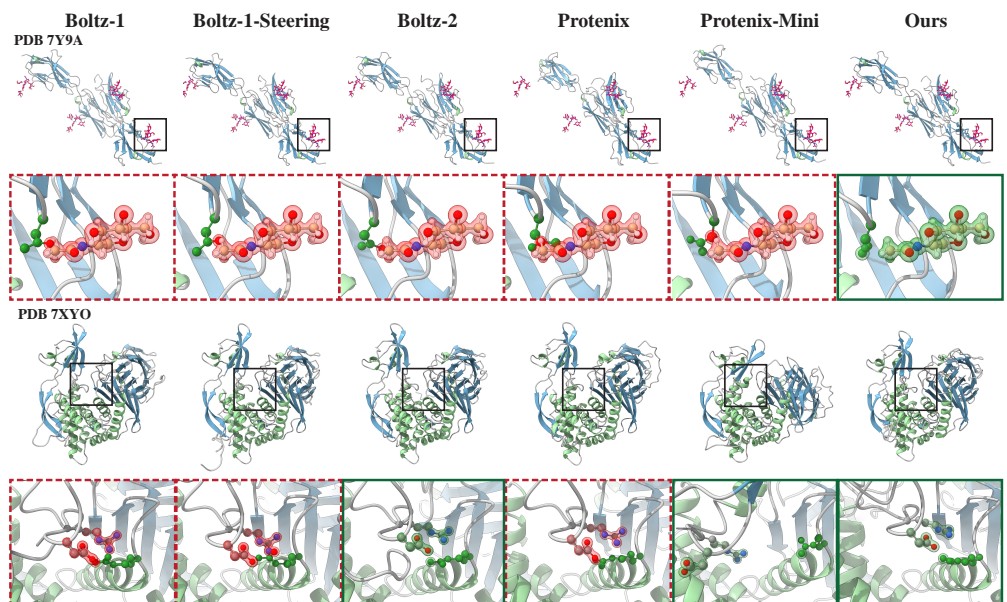

Figure 4: **Qualitative comparison with baseline methods.** The red color highlights physically invalid predictions, such as atomic clashes. Our approach consistently guarantees physical validity. The first example demonstrates a ligand-protein steric clash, where the ligand atoms physically penetrate the receptor surface. This corresponds to constraint 1 (Steric Clash) in Fig. 11. The second example demonstrates an inter-chain overlap, where atoms from Chain A and Chain B occupy the same space. This corresponds to constraint 6 (Overlapping Chains) in Fig. 11.

more iterations to achieve a comparable reduction. This gap reflects the algorithms: Gauss-Seidel projection is of quasi-second order and thus converges faster than first-order gradient descent.

**Wall-clock Runtime Comparison.** To assess practical speed, we measured the wall-clock inference time (including diffusion and physical validity enforcement) on CASP15. The results, binned by atom count, are shown in Fig. 5 (right). Our 2-step predictor with Gauss-Seidel projection is consistently the fastest. It achieves a median speedup of $\sim 9.4\times$ over the 200-step Boltz-1 baseline and $\sim 9.5\times$ over Protenix. Compared with Protenix-Mini with 5 denoising steps, our method achieves $\sim 2.3\times$ speedup. Compared with Boltz-1-Steering, the speedup is $23-46\times$. The gains stem from the reduction in denoising steps and our GPU implementation of the projection module.

## 5.5 ABLATION STUDY

To isolate the impact of our differentiable Gauss-Seidel (GS) projection, we conduct an ablation study on CASP15 using the 200-step Boltz-1 model as a reference. We compare four 2-step sampling variants, presented in the order reported in the inset table: (i) a vanilla 2-step sampler with no mechanism for physical validity; (ii) adding gradient descent (GD) guidance at inference; (iii) adding GS projection at inference (only as a post-processing) without fine-tuning; (iv) finetuning with GS projection and also applying it at inference (our full method).

Moving from the 200-step baseline to a 2-step sampler without guidance reduces both accuracy and physical validity. Adding gradient descent guidance at inference improves validity but fails to recover the structural accuracy. Using Gauss-Seidel only as post-processing guarantees validity, but at the cost of slightly degrading

|  |  | Complex LDDT | Physical Validity |
|---|---|---|---|
|  | 200-step | 0.6406 | 0.6545 |
| (i) | + 2-step (no guidance, no finetune) | 0.5920 | 0.6327 |
| (ii) | + GD at inference, no finetune | 0.5942 | 0.7345 |
| (iii) | + GS at inference, no finetune | 0.5889 | **1.00** |
| (iv) | + GS at inference, finetune | **0.6321** | **1.00** |

the accuracy. Our full setting closes the gap to the 200-step baseline with two denoising steps and is physically valid. These results show that while Gauss-Seidel projection is necessary to enforce validity, making it differentiable is critical for high structural accuracy in the few-step regime.

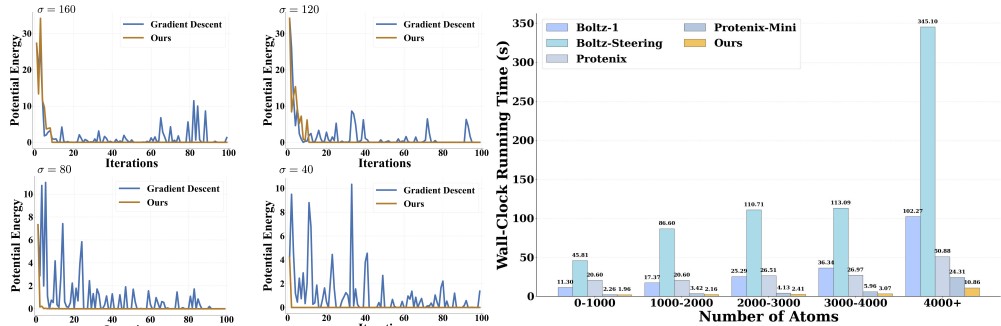

Figure 5: **Convergence and runtime.** Left: Potential energy vs. iteration on PDB 8X51 under four noise levels ($\sigma \in \{160, 120, 80, 40\}$). Gauss-Seidel projection (orange) converges to tolerance within 20 iterations; gradient descent guidance (blue) oscillates and decays slowly. Right: Wall-clock inference time by atom-count bin on CASP15. Ours is $\sim10\times$ faster than 200-step baselines and $\sim2.3\times$ faster than Protenix-Mini, while maintaining validity. Runtime includes diffusion and validity module; measured on the same hardware.

Table 2: **Quantitative results on 6 datasets (Part II).** Dark green cells denote the best results among the few-step methods. Light green cells denote the best results among the 200-step methods.

| | Method / Metric | # Denoise Steps | Complex LDDT | Prot-Prot iLDDT | DNA-Prot iLDDT | RNA-Prot iLDDT | TM-score | Physical Validity |
|---|---|---|---|---|---|---|---|---|
| **AF3-AB** | Boltz-1 | 200 | 0.80±0.02 | 0.12±0.05 | – | – | 0.61±0.06 | 0.00±0.00 |
| | Protenix | 200 | 0.80±0.03 | 0.23±0.06 | – | – | 0.64±0.06 | 0.01±0.01 |
| | Boltz-1-Steering | 200 | 0.79±0.03 | 0.13±0.06 | – | – | 0.62±0.07 | 0.89±0.11 |
| | Boltz-2 | 200 | 0.87±0.03 | 0.55±0.10 | – | – | 0.72±0.07 | 0.89±0.11 |
| | Protenix-Mini | 10 | 0.77±0.03 | 0.11±0.04 | – | – | 0.60±0.06 | 0.00±0.00 |
| | Protenix-Mini | 5 | 0.77±0.03 | 0.11±0.04 | – | – | 0.59±0.06 | 0.00±0.00 |
| | Ours | 2 | 0.78±0.02 | 0.14±0.05 | – | – | 0.60±0.06 | 1.00±0.00 |
| **dsDNA** | Boltz-1 | 200 | 0.85±0.04 | – | 0.76±0.07 | – | 0.89±0.06 | 0.03±0.03 |
| | Protenix | 200 | 0.86±0.04 | – | 0.78±0.08 | – | 0.89±0.06 | 0.04±0.03 |
| | Boltz-1-Steering | 200 | 0.85±0.04 | – | 0.77±0.07 | – | 0.90±0.06 | 1.00±0.00 |
| | Boltz-2 | 200 | 0.94±0.02 | – | 0.91±0.05 | – | 0.93±0.06 | 1.00±0.00 |
| | Protenix-Mini | 10 | 0.82±0.05 | – | 0.71±0.09 | – | 0.85±0.07 | 0.18±0.10 |
| | Protenix-Mini | 5 | 0.81±0.05 | – | 0.71±0.09 | – | 0.85±0.07 | 0.24±0.09 |
| | Ours | 2 | 0.80±0.04 | – | 0.71±0.07 | – | 0.88±0.06 | 1.00±0.00 |
| **RNA-Protein** | Boltz-1 | 200 | 0.74±0.04 | – | – | 0.30±0.07 | 0.85±0.11 | 0.00±0.00 |
| | Protenix | 200 | 0.75±0.05 | – | – | 0.34±0.09 | 0.82±0.12 | 0.02±0.02 |
| | Boltz-1-Steering | 200 | 0.73±0.05 | – | – | 0.30±0.07 | 0.83±0.11 | 0.96±0.04 |
| | Boltz-2 | 200 | 0.88±0.05 | – | – | 0.73±0.11 | 0.88±0.11 | 0.95±0.05 |
| | Protenix-Mini | 10 | 0.73±0.05 | – | – | 0.38±0.10 | 0.80±0.13 | 0.02±0.02 |
| | Protenix-Mini | 5 | 0.72±0.05 | – | – | 0.36±0.10 | 0.79±0.13 | 0.05±0.05 |
| | Ours | 2 | 0.72±0.05 | – | – | 0.32±0.06 | 0.82±0.11 | 1.00±0.00 |

# 6 CONCLUSION

In this work, we propose a differentiable Gauss-Seidel projection module that enforces physical validity during both training and inference for all-atom biomolecular interaction modeling. Framed as a constrained optimization, the module projects the coordinates produced by the diffusion module to configurations without violating physical validity. Finetuning the diffusion module with this projection achieves physically valid and structurally accurate outputs with only 2-step denoising, delivering $\sim 10\times$ faster inference than baselines while maintaining competitive accuracy.

**Limitations and Future Work.** While our projection module substantially reduces the number of denoising steps, single-step inference remains out of reach. Future work will focus on achieving it by combining our projection module with one-step diffusion training techniques.

**Reproducibility Statement.** To ensure the reproducibility of our work, we provide detailed experimental settings in Sec. 5.1, the pseudocode for the Forward Solver and Backward Pass of the

Gauss-Seidel projection module in Sec. 4.2 and Sec. 4.3, the pseudocode for training and inference algorithms in Appendix H, the details on the baselines and the evaluation protocol in Appendix M, and the complete formulations of the physical validity constraints in Appendix L.

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

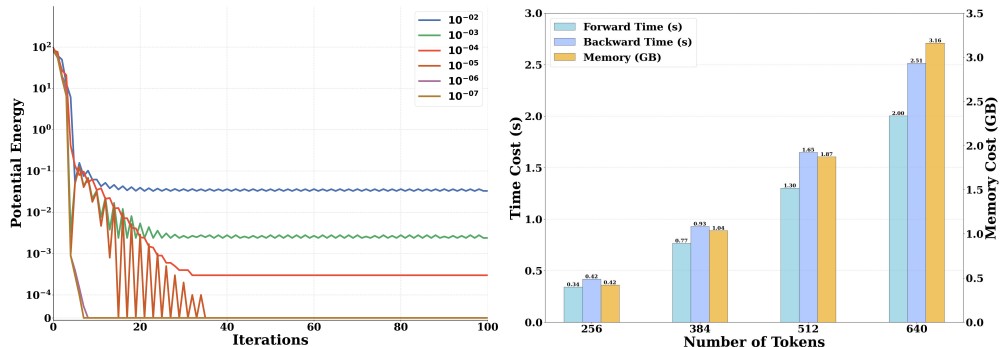

Figure 6: **Convergence and computational cost analysis of the Gauss-Seidel projection module.** Left: Potential energy plotted over iterations for $\alpha$ values ranging from $10^{-2}$ to $10^{-7}$. Large values ($>10^{-5}$) result in persistent oscillations without full convergence. An intermediate value ($\alpha=10^{-5}$) exhibits transient oscillations before reaching zero potential, while smaller values ($10^{-6}, 10^{-7}$) ensure rapid and stable convergence, indicating robust constraint satisfaction. Right: Computational benchmarks showing forward/backward time costs and additional memory overhead across varying token counts (256, 384, 512, and 640).

## A    ADDITIONAL QUALITATIVE RESULTS

Fig. 9 shows more qualitative comparisons with baselines. We also visualize full protein structures predicted by our method and compare them with the ground truth from six datasets, shown in Fig. 10. The results demonstrate that our model accurately predicts physically valid structures.

## B    UNIQUENESS OF THE GAUSS-SEIDEL SOLUTION

The Gauss-Seidel scheme guarantees uniqueness for linearized subproblems. Theoretically, while ordering and initialization govern the convergence speed of the Gauss-Seidel scheme, they do not influence the solution's uniqueness (Golub & Van Loan, 2013).

We investigated this empirically on the PoseBusters dataset by performing five independent projection runs for each denoised structure. The computed pairwise RMSD among outputs was 0.0000 $\pm$ 0.0000 Å. This confirms that, in practice, the projection module behaves as a stable mapping, reliably converging to a numerically identical solution.

## C    PRACTICAL CHOICE OF $\alpha$ AND ITS SENSITIVITY OF CONVERGENCE

To assess how the parameter $\alpha$ influences the convergence of the Gauss-Seidel projection, we conducted an ablation study on the protein PDB 7WUY, evaluating both convergence behavior and predicted accuracy across six values ($\alpha \in \{10^{-2}, \ldots, 10^{-7}\}$). As illustrated in Fig. 6, we monitored the potential energy after each iteration and observed two distinct behaviors: For large $\alpha$ ($10^{-2}, 10^{-3}, 10^{-4}$), it fails to converge to a physically valid configuration. While the potential energy decreases initially, it exhibits persistent oscillations rather than converging to zero. This indicates that the constraint penalties are not enforced strongly enough to resolve physical violations. For small $\alpha$ ($\leq 10^{-5}$), at $\alpha=10^{-5}$, the energy oscillates transiently before eventually converging to zero, confirming that all constraints are resolved. For smaller values ($\alpha=10^{-6}, 10^{-7}$), the projection consistently and rapidly converges to a valid state (zero potential energy), demonstrating fast and stable constraint satisfaction. These results confirm that selecting $\alpha \leq 10^{-6}$ is critical for ensuring the projection reliably maps to a physically valid configuration.

## D    COMPARISONS WITH DIFFERENT DENOISING STEP BUDGETS

We conducted experiments comparing Boltz-1, Boltz-2, and our method (built on Boltz-1) across denoising step budgets of 2, 5, 10, and 200 on the CASP15 dataset. As shown in Fig. 7, our method

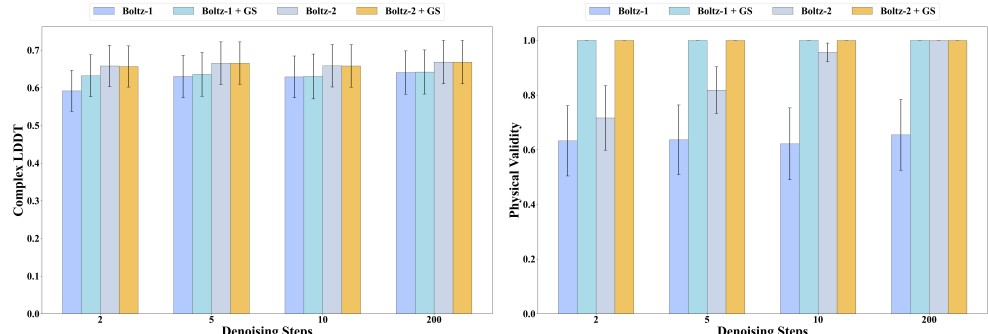

Figure 7: **Comparison of Boltz-1, Boltz-2, and our method across 2-, 5-, 10-, and 200-step budgets.** Our method consistently outperforms the Boltz-1 baseline while maintaining strong physical validity. Although Boltz-2 achieves higher Complex-LDDT scores, integrating our module into the Boltz-2 preserves this accuracy while significantly boosting physical validity at low step budgets.

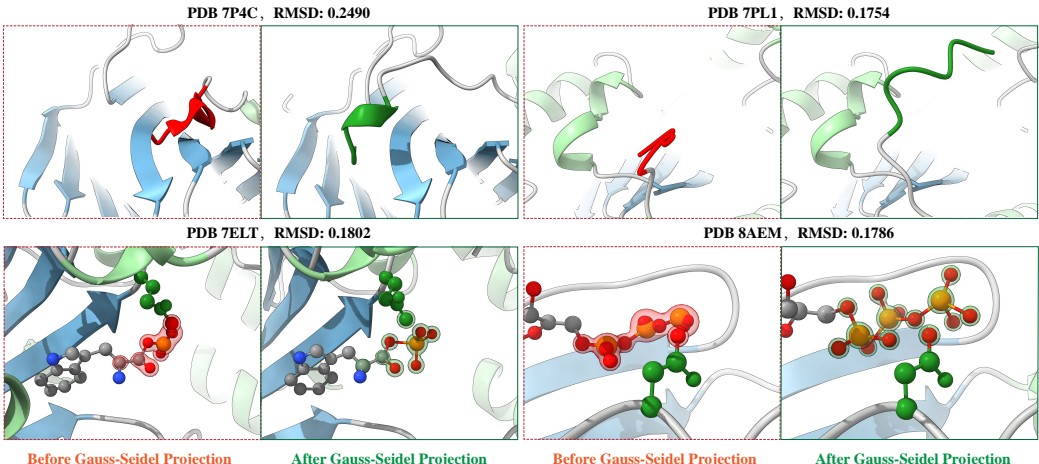

Figure 8: **Visualization of structural corrections by Guass-Seidel projection.** The projection effectively resolves steric violations and restores physically valid geometries. We showcase four representative cases from the PoseBusters dataset. PDB 7P4C: A distorted $\alpha$-helix clashing with a neighboring strand is corrected. PDB 7PL1: A self-entangled coil is disentangled, resolving clashes. PDB 7ELT: Ligand-sidechain interactions and invalid internal ligand geometries are resolved. PDB 8AEM: Ligand-related collisions are removed. For each case, RMSD values were computed to quantify the structural refinement.

demonstrates consistent improvement over the Boltz-1 baseline, outperforming it in terms of both structural accuracy (Complex-LDDT) and physical validity across all step budgets.

We acknowledge that our current implementation does not surpass the performance of Boltz-2. We attribute this to the underlying base model: our method is fine-tuned using the Boltz-1 architecture, weights, and dataset. In contrast, Boltz-2 benefits from significant architectural advancements (e.g., the affinity prediction module) and a much larger training dataset particularly focused on protein complex interactions. This advantage is evident in the interface metrics where Boltz-2 dominates.

However, our Gauss-Seidel projection module is theoretically agnostic to the base model. To validate this potential without the computational cost of full fine-tuning, we applied our module as a post-processing step on top of Boltz-2. Results in Fig. 7 show that this hybrid approach effectively ensures 100% physical validity: it resolves invalid outputs observed in Boltz-2 at low step counts (<10) while maintaining the high structural accuracy of the original model. We thus treat our method as a general-purpose physical validity enforcement module that can be flexibly integrated into various diffusion-based structure prediction frameworks.

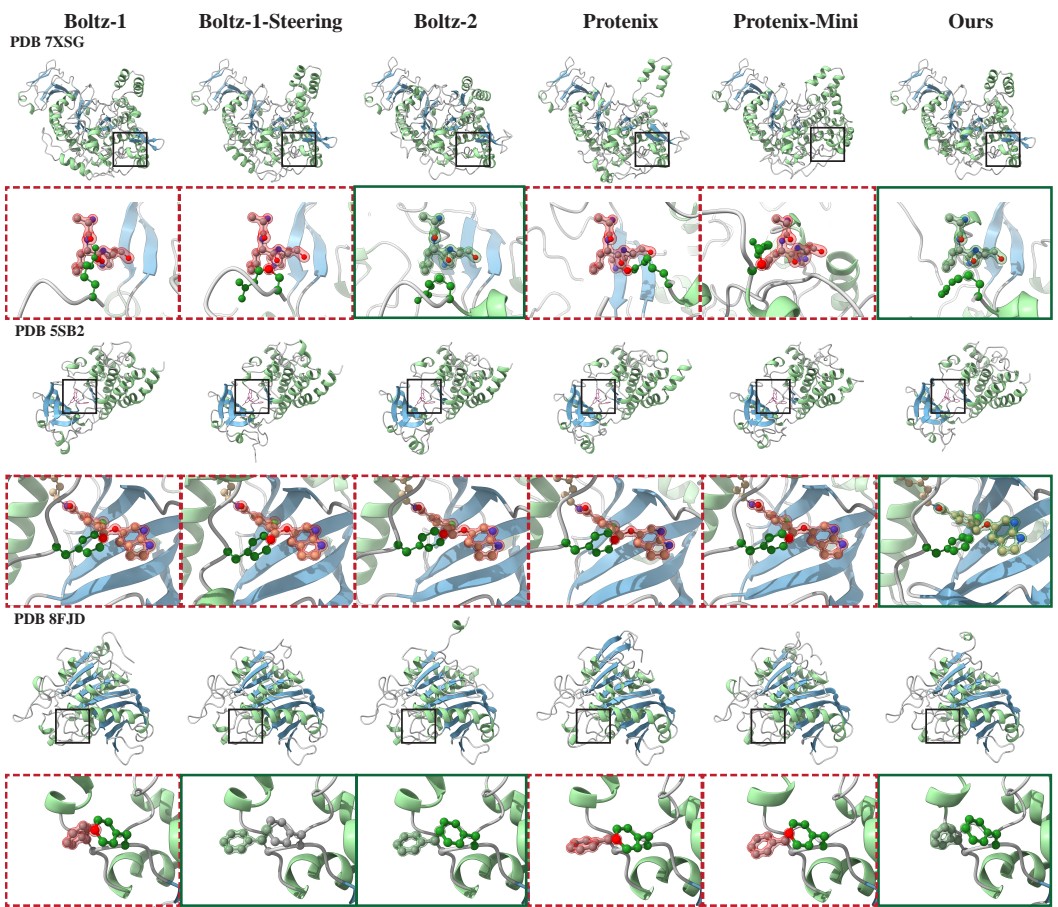

Figure 9: **Additional qualitative comparisons.** The red color highlights physically invalid structural predictions, such as side chain clashes (PDB 7XSG and PDB 8FJD) and ligand-related clashes (PDB 5SB2). Our approach consistently enforces physical validity.

# E    COMPUTATIONAL COST OF THE GAUSS-SEIDEL PROJECTION MODULE

We systematically evaluated the computational overhead of incorporating the Gauss-Seidel projection module during both training and inference. To provide a canonical measure of input complexity, we benchmarked the module on the CASP15 dataset, binning samples by token count (256, 384, 512, and 640). We reported the forward and backward wall-clock runtimes and additional memory usage on an NVIDIA A100 GPU.

As illustrated in Fig. 6, both forward and backward passes are completed within a matter of seconds, and the additional memory overhead remains below 3.5 GB, even for the largest token bins. This high efficiency is achieved through our specialized implementation, which leverages custom CUDA kernels optimized for both the forward and backward processes.

Table 3: **L-RMSD < 2 Å on the PoseBusters dataset computed by PXMeter (Ma et al., 2025).** Dark green cells denote the best results among the few-step methods. Light green cells denote the best results among the 200-step methods.

| Dataset \ Method | AlphaFold3 | Boltz-1 | Protenix | Boltz-1-Steering | Boltz-2 | Protenix-Mini 10-step | Protenix-Mini 5-step | Ours |
|---|---|---|---|---|---|---|---|---|
| PoseBusters | 0.86±0.04 | 0.71±0.05 | 0.80±0.05 | 0.71±0.05 | 0.91±0.03 | 0.76±0.05 | 0.72±0.05 | 0.72±0.05 |

## F    CLARIFICATION ON DOCKING-RELATED METRIC COMPUTATION

In our primary evaluation, we calculated docking-related metrics (DockQ $> 0.23$, and L-RMSD $< 2$Å) using OpenStructure (Biasini et al., 2013), strictly adhering to the protocol defined in the original Boltz-1 paper (Wohlwend et al., 2024). We attribute the discrepancies between our reported values and those in the Protenix technical report (Team et al., 2025) to their use of a different evaluation tool, PXMeter (Ma et al., 2025). To investigate this, we performed an evaluation of the L-RMSD $< 2$Å metric using PXMeter on the PoseBusters dataset for all methods (detailed in Table 3). This re-evaluation confirmed two key points: 1) The re-calculated performance of the Protenix baseline aligns closely with the figures in their original report, validating the source of the numerical difference. 2) While the absolute values depending on the tool used, the *relative ranking* among the baselines remains unchanged. These results indicate that our comparative analysis is informative.

## G    ANALYSIS OF STRUCTURAL PERTURBATIONS INTRODUCED BY GAUSS-SEIDE PROJECTION

We quantified the structural changes introduced by Gauss-Seidel projection by computing the RMSD between pre- and post-projection structures on the PoseBusters dataset. The observed average RMSD was $0.1865 \pm 0.0219$ Å, confirming that the projection step introduces atomic position changes sufficient to enforce physical validity. Fig. 8 visualizes four representative cases where noticeable improvements occurred. For PDB 7P4C, the projection restored a distorted $\alpha$-helix that was originally clashing with a neighboring strand. In PDB 7PL1, a self-entanglement within a coil region was successfully disentangled. Similarly, for PDB 7ELT and PDB 8AEM, the module resolved ligand-sidechain and intra-ligand steric clashes. These examples demonstrate that the Gauss-Seidel projection effectively enforces physical validity while preserving the structural prediction of the generative model.

## H    TRAINING AND INFERENCE ALGORITHMS

We provide pseudo-code for the training and inference algorithms of our model, shown in Algorithms 3 and 4. The values of hyperparameters shown in both algorithms follow (Gong et al., 2025).

---

**Algorithm 3** Training with Gauss-Seidel Projection

---

**Require:** Training dataset consisting of feature-coordinate pairs, $\mathcal{D} = \{(f^*, \mathbf{x}^{\text{target}})\}$, optimizer $\mathcal{O}$, number of epochs $E$, number of cycles $N_{\text{cycle}}$, noise $c_0$, noise scale $\lambda$

1: **for** epoch $= 1, \ldots, E$ **do**
2:     **for** each batch $(\{f^*\}, \{\mathbf{x}^{\text{target}}\}) \in \mathcal{D}$ **do**
3:         $\{s_i^{\text{inputs}}\} \leftarrow$ ATOM_ATTENTION_ENCODER$(\{f^*\})$
4:         Initialize $\{s_i\}, \{z_{ij}\}$ from $\{s_i^{\text{inputs}}\}$
5:         **for** $c \in [1, \ldots, N_{\text{cycle}}]$ **do**
6:             $\{z_{ij}\} \leftarrow$ MSA_MODULE$(\{f_S^{\text{msa}}\}, \{z_{ij}\}, \{s_i^{\text{inputs}}\})$
7:             $\{s_i\}, \{z_{ij}\} \leftarrow$ PAIRFORMER_MODULE$(\{s_i\}, \{z_{ij}\})$
8:         **end for**
9:         Sample noise: $\xi \sim \mathcal{N}(\mathbf{0}, I_{N \times 3})$
10:         Construct noisy input: $\mathbf{x}^{\text{noisy}} = c_0 \cdot \mathbf{x}^{\text{target}} + \lambda \cdot \xi$
11:         $\{\hat{\mathbf{x}}\} =$ DIFFUSION_MODULE$(\{\mathbf{x}^{\text{noisy}}\}, c_0, \{f^*\}, \{s_i^{\text{inputs}}\}, \{s_i\}, \{z_{ij}\})$
12:         $\{\mathbf{x}^{\text{pred}}\} =$ GAUSS-SEIDEL_PROJECTION_MODULE$(\{\hat{\mathbf{x}}\})$
13:         Compute loss: $\{\mathcal{L}\} =$ DIFFUSION_LOSS$(\{\mathbf{x}^{\text{pred}}\}, \{\mathbf{x}^{\text{target}}\})$
14:         $\mathcal{O}$.zero_grad()
15:         Backpropagate: $\mathcal{L}$.backward()
16:         Update parameters: $\mathcal{O}$.step()
17:     **end for**
18: **end for**

---

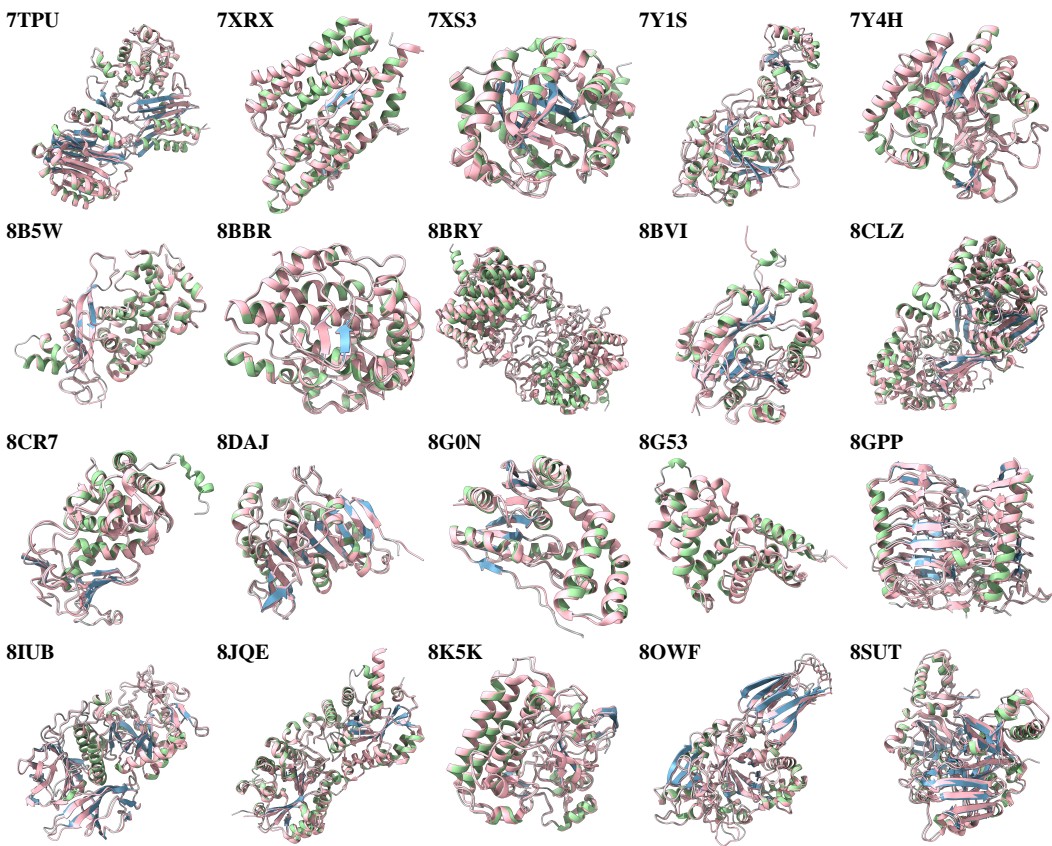

Figure 10: **Visual comparison with the ground truth.** We visualize our predicted protein structures together with their corresponding ground truth (pink colored) across a diverse set of PDB entries. Each protein's name is indicated in the top-left corner. Our method shows consistently close agreement with the ground truth, capturing the overall structural features with high fidelity.

## I   PROOF FOR THE CONVERGENCE OF LINEARIZATION

**Theorem I.1** *Consider the nonlinear coupled system*

$$
\begin{aligned}
\big(\mathbf{x} - \hat{\mathbf{x}}\big) - \nabla\mathbf{C}(\mathbf{x})^{\top}\boldsymbol{\lambda}(\mathbf{x}) &= \mathbf{0}, \\
\mathbf{C}(\mathbf{x}) + \boldsymbol{\alpha}\,\boldsymbol{\lambda}(\mathbf{x}) &= \mathbf{0},
\end{aligned}
\qquad \boldsymbol{\alpha} \succ \mathbf{0},
$$

*where* $\mathbf{C} : \mathbb{R}^{N \times 3} \to \mathbb{R}^{m}$ *is twice continuously differentiable. Let the linearized iteration be*

$$
\begin{bmatrix}
\mathbf{I} & -\nabla\mathbf{C}(\mathbf{x}^{(n)})^{\top} \\
\nabla\mathbf{C}(\mathbf{x}^{(n)}) & \boldsymbol{\alpha}
\end{bmatrix}
\begin{bmatrix}
\Delta\mathbf{x} \\
\Delta\boldsymbol{\lambda}
\end{bmatrix}
= -
\begin{bmatrix}
\mathbf{0} \\
\mathbf{C}(\mathbf{x}^{(n)}) + \boldsymbol{\alpha}\,\boldsymbol{\lambda}^{(n)}
\end{bmatrix},
\tag{9}
$$

*with updates* $\mathbf{x}^{(n+1)} = \mathbf{x}^{(n)} + \Delta\mathbf{x}$ *and* $\boldsymbol{\lambda}^{(n+1)} = \boldsymbol{\lambda}^{(n)} + \Delta\boldsymbol{\lambda}$, *and initial values* $\mathbf{x}^{(0)} = \hat{\mathbf{x}}$, $\boldsymbol{\lambda}^{(0)} = \mathbf{0}$.

*Then the linearized iterates* $(\mathbf{x}^{(n)}, \boldsymbol{\lambda}^{(n)})$ *converge to a solution* $(\mathbf{x}^{*}, \boldsymbol{\lambda}^{*})$ *of the nonlinear system.*

*Proof.* Define

$$
\mathbf{g}(\mathbf{x}, \boldsymbol{\lambda}) = (\mathbf{x} - \hat{\mathbf{x}}) - \nabla\mathbf{C}(\mathbf{x})^{\top}\boldsymbol{\lambda}, \qquad \mathbf{h}(\mathbf{x}, \boldsymbol{\lambda}) = \mathbf{C}(\mathbf{x}) + \boldsymbol{\alpha}\boldsymbol{\lambda},
$$

so that the update of Eq. 9 is equivalent to solving the nonlinear system $\{\mathbf{g} = \mathbf{0},\ \mathbf{h} = \mathbf{0}\}$. Linearizing this system at a given point $(\mathbf{x}^{(i)}, \boldsymbol{\lambda}^{(i)})$ yields the Newton subproblem

$$
\begin{bmatrix}
\mathbf{K} & -\nabla\mathbf{C}(\mathbf{x}^{(i)})^{\top} \\
\nabla\mathbf{C}(\mathbf{x}^{(i)}) & \boldsymbol{\alpha}
\end{bmatrix}
\begin{bmatrix}
\Delta\mathbf{x} \\
\Delta\boldsymbol{\lambda}
\end{bmatrix}
= -
\begin{bmatrix}
\mathbf{g}(\mathbf{x}^{(i)}, \boldsymbol{\lambda}^{(i)}) \\
\mathbf{h}(\mathbf{x}^{(i)}, \boldsymbol{\lambda}^{(i)})
\end{bmatrix},
$$

---

**Algorithm 4** Inference with Gauss-Seidel Projection

---

**Require:** Feature $\{f^*\}$, number of cycles $N_{\text{cycle}}$, noise schedule $[c_0, c_1, c_2]$, $\gamma_0$, $\gamma_{\min}$, noise scale $\lambda$, step scale $\eta$

1: $\{s_i^{\text{inputs}}\} \leftarrow$ ATOM_ATTENTION_ENCODER$(\{f^*\})$
2: Initialize $\{s_i\}, \{z_{ij}\}$ from $\{s_i^{\text{inputs}}\}$
3: **for** $c \in [1, \ldots, N_{\text{cycle}}]$ **do**
4: $\quad \{z_{ij}\} \leftarrow$ MSA_MODULE$(\{f_S^{\text{msa}}\}, \{z_{ij}\}, \{s_i^{\text{inputs}}\})$
5: $\quad \{s_i\}, \{z_{ij}\} \leftarrow$ PAIRFORMER_MODULE$(\{s_i\}, \{z_{ij}\})$
6: **end for**
7: $\mathbf{x} \sim c_0 \cdot \mathcal{N}(\mathbf{0}, I_{N \times 3})$
8: **for** $c_\tau \in [c_1, c_2]$ **do** $\qquad\qquad\qquad\qquad\qquad\qquad\qquad$ ▷ 2-step diffusion
9: $\quad \{\mathbf{x}\} \leftarrow$ CENTRE_RANDOM_AUGMENTATION$(\{\mathbf{x}\})$
10: $\quad \gamma = \gamma_0$ if $c_\tau > \gamma_{\min}$ else $0$
11: $\quad \hat{t} = c_{\tau-1}(\gamma + 1)$
12: $\quad \xi = \lambda\sqrt{\hat{t}^2 - c_{\tau-1}^2} \cdot \mathcal{N}(\mathbf{0}, I_{N \times 3})$
13: $\quad \mathbf{x}^{\text{noisy}} = \mathbf{x} + \xi$
14: $\quad \{\mathbf{x}^{\text{denoised}}\} =$ DIFFUSION_MODULE$(\{\mathbf{x}^{\text{noisy}}\}, \hat{t}, \{f^*\}, \{s_i^{\text{inputs}}\}, \{s_i\}, \{z_{ij}\})$
15: $\quad \delta = (\mathbf{x} - \mathbf{x}^{\text{denoised}})/\hat{t}$
16: $\quad \mathrm{d}t = c_\tau - \hat{t}$
17: $\quad \hat{\mathbf{x}} = \mathbf{x}^{\text{noisy}} + \eta \cdot \mathrm{d}t \cdot \delta_l$
18: $\quad \mathbf{x} \leftarrow \hat{\mathbf{x}}$
19: **end for**
20: $\{\mathbf{x}_{\text{proj}}\} =$ GAUSS-SEIDEL_PROJECTION_MODULE$(\{\hat{\mathbf{x}}\})$
21: **return** $\{\mathbf{x}_{\text{proj}}\}$

---

with $\mathbf{K} = \mathbf{I} - \sum_j \lambda_j \nabla^2 \mathbf{C}_j(\mathbf{x}^{(i)})$. We thus solve, at each iteration, a linear system obtained by Newton linearization of the coupled equations. By Newton's local convergence theorem (Sauer, 2011), the iterates converge quadratically given the Jacobian at the solution $(\mathbf{x}^*, \boldsymbol{\lambda}^*)$ is non-singular and the initialization lies in its basin of attraction. Taking the Schur complement (Zhang, 2005) with respect to $\boldsymbol{\alpha} \succ \mathbf{0}$ shows that the non-singularity of the Jacobian is equivalent to the invertibility of

$$\mathbf{S} = \left(\mathbf{I} - \sum_j \lambda_j^* \nabla^2 \mathbf{C}_j(\mathbf{x}^*)\right) - (-\nabla\mathbf{C}(\mathbf{x}^*)^\top)\boldsymbol{\alpha}^{-1}\nabla\mathbf{C}(\mathbf{x}^*).$$

For sufficiently small yet positive definite $\boldsymbol{\alpha}$, the term $\nabla\mathbf{C}(\mathbf{x}^*)^\top \boldsymbol{\alpha}^{-1} \nabla\mathbf{C}(\mathbf{x}^*)$ dominates and $\mathbf{S}$ becomes strictly positive definite under standard regularity (full row rank of $\nabla\mathbf{C}$ at $\mathbf{x}^*$), hence invertible. Consequently, the Jacobian is non-singular at the solution, and Newton's method is guaranteed to converge locally. In practice, the predictor's output $\mathbf{x}^{(0)} = \hat{\mathbf{x}}$ is close to $\mathbf{x}^*$, so the initialization within the convergence neighborhood.

We then introduce two controlled approximations to avoid forming the Hessian terms in $\mathbf{K}$. First, we replace $\mathbf{K}$ by the identity $\mathbf{I}$ (i.e., a quasi-Newton approximation), which omits the Hessian terms and introduces a local error of order $O(\|\Delta\mathbf{x}\|^2)$ while retaining a positive-definite $\mathbf{S}$. This does not affect the global error or the solution of the fixed-point iteration (Macklin & Müller, 2016). Second, we note that $\mathbf{g}(\mathbf{x}^{(0)}, \boldsymbol{\lambda}^{(0)}) = \mathbf{0}$ and that $\mathbf{g}(\mathbf{x}^{(i)}, \boldsymbol{\lambda}^{(i)})$ remains small when constraint gradients vary slowly; accordingly, we set $\mathbf{g}(\mathbf{x}^{(i)}, \boldsymbol{\lambda}^{(i)}) \approx \mathbf{0}$ (Goldenthal et al., 2007; Macklin & Müller, 2016). Under these approximations, the linear subproblem reduces to Eq. 9.

## J PROOF FOR THE CONVERGENCE AND CONSTRAINT SATISFACTION OF GAUSS-SEIDEL PROJECTION

**Theorem J.1** *Consider the linear system*

$$\left(\nabla\mathbf{C}(\mathbf{x}^{(n)})\nabla\mathbf{C}(\mathbf{x}^{(n)})^\top + \boldsymbol{\alpha}\right)\Delta\boldsymbol{\lambda} = -\mathbf{C}(\mathbf{x}^{(n)}) - \boldsymbol{\alpha}\boldsymbol{\lambda}^{(n)}, \quad \boldsymbol{\alpha} > \mathbf{0}, \tag{10}$$

*with each Gauss-Seidel update*

$$\Delta\boldsymbol{\lambda}_j = \frac{-\mathbf{C}_j(\mathbf{x}^{(n)}) - \boldsymbol{\alpha}_j\boldsymbol{\lambda}_j^{(n)}}{\nabla\mathbf{C}_j(\mathbf{x}^{(n)})\nabla\mathbf{C}_j(\mathbf{x}^{(n)})^\top + \boldsymbol{\alpha}_j}, \qquad j = 1, ..., m, \tag{11}$$

*at the $n$-th Gauss-Seidel sweep. After a sufficient number of Gauss-Seidel sweeps and for sufficiently small $\boldsymbol{\alpha}$, the iterates converge in a quasi-second-order manner to the unique solution $\Delta\boldsymbol{\lambda}^*$. Moreover, each individual update monotonically reduces the residual error and requires only $O(1)$ work, so a full sweep over all $m$ constraints costs $O(m)$.*

*Proof.* Define

$$\mathbf{A} = \nabla\mathbf{C}(\mathbf{x}^{(n)})\nabla\mathbf{C}(\mathbf{x}^{(n)})^\top + \boldsymbol{\alpha}, \quad \mathbf{b} = -\mathbf{C}(\mathbf{x}^{(n)}) - \boldsymbol{\alpha}\boldsymbol{\lambda}^{(n)}.$$

The solution of Eq. 10 corresponds exactly to the minimizer of the quadratic optimization problem:

$$\min_{\Delta\boldsymbol{\lambda}} f(\Delta\boldsymbol{\lambda}) = \frac{1}{2}\Delta\boldsymbol{\lambda}^\top \mathbf{A}\, \Delta\boldsymbol{\lambda} - \mathbf{b}^\top\Delta\boldsymbol{\lambda}. \tag{12}$$

With $\boldsymbol{\alpha} \succ \mathbf{0}$, $\mathbf{A}$ is symmetric positive definite (SPD). Hence, Eq. 12 is convex, and Eq. 10 has a unique solution.

A cyclic Gauss-Seidel sweep on Eq. 10 is exactly cyclic coordinate descent on Eq. 12: each update minimizes $f$ w.r.t. a single coordinate $\Delta\lambda_j$ while holding the others fixed, yielding the closed form in Eq. 11. Because $f$ is strongly convex and quadratic, each coordinate step strictly decreases $f$ unless already optimal, and the sequence $\{f(\Delta\boldsymbol{\lambda}^{(k)})\}$ is monotonically decreasing. For SPD systems, both classical matrix-splitting analysis and coordinate-descent theory guarantee linear convergence of cyclic Gauss-Seidel to the unique minimizer (Varga, 1962; Saad, 2003; Wright, 2015). In particular, there exists $\rho \in (0, 1)$ such that $\|\Delta\boldsymbol{\lambda}^{(k)} - \Delta\boldsymbol{\lambda}^*\|_{\mathbf{A}} \leq \rho^k\|\Delta\boldsymbol{\lambda}^{(0)} - \Delta\boldsymbol{\lambda}^*\|_{\mathbf{A}}$. Each coordinate update accesses only the atoms affected by the current constraint, so it is $O(1)$. A full sweep is thus $O(m)$.

The proof of Theorem I.1 shows that Eq. 10 is a quasi-Newton approximation of the first-order optimality condition Eq. 3. Hence, the convergence speed of the Gauss-Seidel projection is of quasi-second order. With a strict physical tolerance and sufficiently small penalty parameter $\boldsymbol{\alpha}$, the final iterate satisfies the optimality conditions of the penalized projection, meaning that in practice it realizes a hard-constrained projection (all constraints are strictly satisfied) (Boyd & Vandenberghe, 2004).

## K  PROOF FOR THE CONVERGENCE OF CONJUGATE GRADIENT

**Theorem K.1** *Consider the adjoint system*

$$\left(\mathbf{H}(\mathbf{x}_{\text{proj}}) + \mathbf{I}\right)^\top \mathbf{z} = \left(\frac{\partial L}{\partial\mathbf{x}_{\text{proj}}}\right)^\top, \tag{13}$$

*where*

$$\mathbf{H}(\mathbf{x}) = \sum_{j=1}^m \alpha_j^{-1}\left(\nabla\mathbf{C}_j(\mathbf{x})\,\nabla\mathbf{C}_j(\mathbf{x})^\top + \mathbf{C}_j(\mathbf{x})\,\nabla^2\mathbf{C}_j(\mathbf{x})\right). \tag{14}$$

*Near feasibility, the conjugate gradient method converges when applied to this system.*

*Proof.* At the projected point $\mathbf{x}_{\text{proj}}$, we have $\mathbf{C}(\mathbf{x}_{\text{proj}}) = 0$. Hence the second-order small term $\mathbf{C}_j(\mathbf{x}_{\text{proj}})\,\nabla^2\mathbf{C}_j(\mathbf{x}_{\text{proj}})$ vanish, and

$$\mathbf{H}(\mathbf{x}_{\text{proj}}) = \sum_{j=1}^m \alpha_j^{-1}\,\nabla\mathbf{C}_j(\mathbf{x}_{\text{proj}})\nabla\mathbf{C}_j(\mathbf{x}_{\text{proj}})^\top.$$

This matrix is SPD, and so is $\mathbf{H}(\mathbf{x}_{\text{proj}}) + \mathbf{I}$. The conjugate gradient method is guaranteed to converge for any linear system with a SPD coefficient matrix (Sauer, 2011), and its convergence rate is governed by the condition number of the matrix. Since the coefficient matrix contains the identity $\mathbf{I}$, the condition number is bounded below by the size of $\mathbf{I}$, i.e., $m$, which ensures that the convergence of conjugate gradient remains stable in the neighborhood of feasibility.

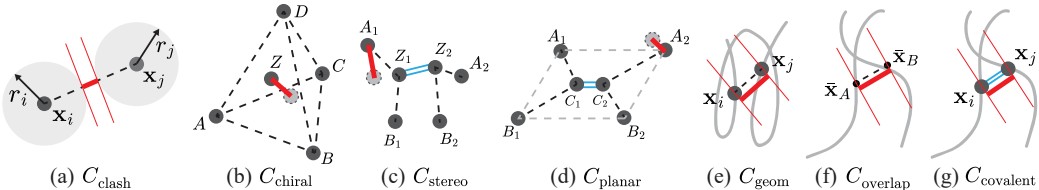

$$(a)\ C_{\text{clash}} \qquad (b)\ C_{\text{chiral}} \qquad (c)\ C_{\text{stereo}} \qquad (d)\ C_{\text{planar}} \qquad (e)\ C_{\text{geom}} \quad (f)\ C_{\text{overlap}} \quad (g)\ C_{\text{covalent}}$$

Figure 11: **Visualization of the physical constraints used in our framework.** Red highlights indicate the structural or geometric elements subject to constraints, such as spatial arrangements and interatomic distances, while blue highlights denote double bonds or covalent linkages between atoms.

## L   DETAILED FORMULATION OF PHYSICAL VALIDITY CONSTRAINTS

Below we present the formalization of each of the physical constraints following Boltz-1-Steering (Wohlwend et al., 2024), which are visualized in Fig. 11.

**(a) Steric Clash.**   To prevent steric clashes, we impose a constraint on the distance between atoms in distinct, non-bonded chains, requiring it to be greater than 0.725 times the sum of their Van der Waals radii:

$$C_{\text{clash}}(\mathbf{x}) = \sum_{(i,j) \in S_{\text{cross chains}}} \max\Big(0.725(r_i + r_j) - \|\mathbf{x}_i - \mathbf{x}_j\|, 0\Big), \tag{15}$$

where $r_i$ denotes the Van der Waals radius of atom $i$.

**(b) Tetrahedral Atom Chirality.**   For a chiral center $Z$ with four substituents ordered by Cahn-Ingold-Prelog (CIP) priority as $A, B, C, D$, the center is designated $R$ if the bonds $(Z - A, Z - B, Z - C)$ form a right-handed system, and $S$ if they form a left-handed system. To ensure that the predicted molecular conformers maintain the correct chirality, we define the constraint based on the improper torsion angles $(X_1, X_2, X_3, Z)$:

$$\begin{aligned}
C_{\text{chiral}}(\mathbf{x}) = &\sum_{(i,j,k,l) \in S_{\text{R chiral sets}}} \max\Big(\frac{\pi}{6} - \text{DihedralAngle}(\mathbf{x}_i, \mathbf{x}_j, \mathbf{x}_k, \mathbf{x}_l), 0\Big) \\
&+ \sum_{(i,j,k,l) \in S_{\text{S chiral sets}}} \max\Big(\frac{\pi}{6} + \text{DihedralAngle}(\mathbf{x}_i, \mathbf{x}_j, \mathbf{x}_k, \mathbf{x}_l), 0\Big).
\end{aligned} \tag{16}$$

**(c) Bond Stereochemistry.**   For a double bond $Z_1 = Z_2$, where $Z_1$ has substituents $A_1, B_1$ and $Z_2$ has substituents $A_2, B_2$ arranged in decreasing Cahn-Ingold-Prelog (CIP) priority, the double bond is said to have $E$ stereochemistry if the higher-priority atoms $A_1$ and $A_2$ lie on opposite sides of the bond, and $Z$ stereochemistry otherwise. To ensure that the predicted molecular conformers maintain the correct stereochemistry, we define the constraint based on the torsion angles $(A_1, Z_1, Z_2, A_2)$ and $(B_1, Z_1, Z_2, B_2)$:

$$\begin{aligned}
C_{\text{stereo}}(\mathbf{x}) = &\sum_{(i,j,k,l) \in S_{\text{E stereo sets}}} \max\Big(\frac{5\pi}{6} - \text{DihedralAngle}(\mathbf{x}_i, \mathbf{x}_j, \mathbf{x}_k, \mathbf{x}_l), 0\Big) \\
&+ \sum_{(i,j,k,l) \in S_{\text{Z stereo sets}}} \max\Big(\text{DihedralAngle}(\mathbf{x}_i, \mathbf{x}_j, \mathbf{x}_k, \mathbf{x}_l) - \frac{\pi}{6}, 0\Big).
\end{aligned} \tag{17}$$

**(d) Planar Double Bonds.**   For planar double bonds $C_1 = C_2$ between carbon atoms, where $C_1$ has substituents $A_1, B_1$ and $C_2$ has substituents $A_2, B_2$, we define a flat-bottom constraint based on the improper torsion angles $(A_1, B_1, C_2, C_1)$ and $(A_2, B_2, C_1, C_2)$ to enforce planarity of the double bond:

$$C_{\text{planar}}(\mathbf{x}) = \sum_{(i,j,k,l) \in S_{\text{trigonal planar sets}}} \max\Big(\text{DihedralAngle}(\mathbf{x}_i, \mathbf{x}_j, \mathbf{x}_k, \mathbf{x}_l) - \frac{\pi}{12}, 0\Big). \tag{18}$$

**(e) Internal Geometry.** To ensure that the model generates ligand conformers with physically realistic distance geometry, we define a flat-bottomed constraint based on the bounds matrices generated by the RDKit package:

$$
\begin{aligned}
C_{\text{geom}}(\mathbf{x}) = &\sum_{(i,j) \in S_{\text{bonds}}} \max(\|\mathbf{x}_i - \mathbf{x}_j\| - 1.2 U_{ij}, 0) + \max(0.8 L_{ij} - \|\mathbf{x}_i - \mathbf{x}_j\|, 0) \\
+ &\sum_{(i,j) \in S_{\text{angles}}} \max(\|\mathbf{x}_i - \mathbf{x}_j\| - 1.2 U_{ij}, 0) + \max(0.8 L_{ij} - \|\mathbf{x}_i - \mathbf{x}_j\|, 0) \\
+ &\sum_{(i,j) \notin S_{\text{bonds}} \cup S_{\text{angles}}} \max(\|\mathbf{x}_i - \mathbf{x}_j\| - 1.2 U_{ij}, 0) + \max(0.8 L_{ij} - \|\mathbf{x}_i - \mathbf{x}_j\|, 0),
\end{aligned}
\tag{19}
$$

where $L$ and $U \in \mathbb{R}^{N \times N}$ denote the lower and upper bounds matrices. For a pair of atoms $(i, j)$, $L_{i,j}$ gives the lower distance bound and $U_{i,j}$ gives the upper distance bound (Buttenschoen et al., 2024).

**(f) Overlapping Chains.** To prevent overlapping chains, we define a constraint based on the distance between the centroids of symmetric chains with more than one atom:

$$
C_{\text{overlap}}(\mathbf{x}) = \sum_{(A,B) \in S_{\text{symmetric chains}}} \max \left( 1.0 - \|\bar{\mathbf{x}}_A - \bar{\mathbf{x}}_B\|, 0 \right),
\tag{20}
$$

where $\bar{\mathbf{x}}_A$ is the centroid of chain $A$.

**(g) Covalently Bonded Chains.** To ensure the model respects covalently bonded chains, we define a constraint to enforce that covalently bonded atoms between separate chains are within 2 Å:

$$
C_{\text{covalent}}(\mathbf{x}) = \sum_{(i,j) \in S_{\text{covalent bonds}}} \max \left( \|\mathbf{x}_i - \mathbf{x}_j\| - 2, 0 \right).
\tag{21}
$$

## M DETAILS ON BASELINES AND EVALUATION PROTOCOL

We compare our method against strong baselines: Boltz-1 (the original model) (Wohlwend et al., 2024), Boltz-1-Steering (Boltz-1 with FK-steering at inference), Boltz-2 (the updated Boltz model) (Passaro et al., 2025), Protenix (Team et al., 2025) and Protenix-Mini (Gong et al., 2025). All baselines use 200-step diffusion sampling, whereas Protenix-Mini uses a 5-step sampling. We evaluate our method and all baselines on six benchmarks: CASP15, Test, PoseBusters, AF3-AB, dsDNA, and RNA-Protein. CASP15 and Test (from Boltz-1 (Wohlwend et al., 2024)) cover diverse assemblies, including protein complexes, nucleic acids, and small molecules. PoseBusters (Buttenschoen et al., 2024) focuses on protein-ligand modeling. AF3-AB (released with AlphaFold3 (Abramson et al., 2024)) contains protein-antibody complexes. dsDNA (Ma et al., 2025) contains entries with two DNA chains and one protein chain. RNA-Protein (Ma et al., 2025) contains entries consisting of one RNA chain and one protein chain.

We compare our method with baselines using standard evaluation metrics, including Complex LDDT, Prot-Prot iLDDT, Lig-Prot iLDDT, DNA-Prot iLDDT, RNA-Prot iLDDT, DockQ > 0.23, L-RMSD < 2 Å, TM-score, and Physical Validity. Complex LDDT measures the overall geometric accuracy throughout the complex, while Prot-Prot iLDDT, Lig-Prot iLDDT, DNA-Prot iLDDT, and RNA-Prot iLDDT focus specifically on the quality of protein-protein, protein-ligand, protein-DNA, and protein-RNA interfaces, respectively. The average DockQ success rate (DockQ > 0.23), defined as the proportion of predictions with DockQ > 0.23, measures the fraction of cases where good protein-protein interactions are correctly predicted. L-RMSD is defined as the proportion of ligands with a pocket-aligned RMSD below 2 Å, a widely adopted measure of molecular docking accuracy. TM-score is a length-normalized measure of structural similarity, widely used to assess protein structure prediction accuracy. Finally, Physical Validity checks whether the predicted structures obey fundamental physical constraints, including steric clash, tetrahedral atomic chirality, bond stereochemistry, planar double bonds, internal geometry, overlapping chains, and covalently bound chains. All evaluation metrics follow those used in Boltz-1 (Wohlwend et al., 2024) and Protenix (Team et al., 2025).

## N    THE USE OF LARGE LANGUAGE MODELS (LLMs)

Consistent with the conference policy on LLM usage, we used LLMs solely for language polishing and improving the clarity of presentation. In particular:

- **Scope of use.** LLMs were used for language polishing (rephrasing sentences, tightening transitions, standardizing terminology). They were *not* used for research ideation, experimental design, data analysis, derivations, algorithmic contributions, code or figure generation, literature search/screening, or drafting technical content beyond author-provided prose.

- **Authorship and responsibility.** All scientific claims, methods, proofs, experiments, and conclusions are authored by the listed authors. We carefully reviewed and verified any text edited with LLM assistance. The authors accept full responsibility for the contents of the paper, including any edited passages, and for ensuring the absence of plagiarism or fabricated material.

- **Attribution and citations.** All references and technical statements were selected, verified, and cited by the authors; LLMs were not used to generate or suggest citations.

- **Data and privacy.** No proprietary, confidential, or personally identifiable information was provided to LLM tools.

