# OpenReview forum: "Physically Valid Biomolecular Interaction Modeling with Gauss-Seidel Projection"
_ICLR.cc/2026/Conference — ICLR 2026 Poster_

### Official Review · Reviewer_iHkC · 2025-10-16

**Soundness:** 3
**Presentation:** 3
**Contribution:** 3
**Rating:** 6
**Confidence:** 3

**Summary:**

This paper addresses the physical validity challenge in generative modeling for protein structures. It proposes a first-order constraint and enforces it during both fine-tuning and inference processes. The constraint is formulated as an optimization problem, which is approximately solved using Gauss-Seidel updates. Additionally, the paper introduces implicit differentiation to enable backpropagation during fine-tuning. Compared to baseline methods, this approach achieves faster inference, requiring only 2-step denoising, while maintaining competitive accuracy.

**Strengths:**

1. The manuscript is well-written, and easy to follow.
2. The performance in terms of inference acceleration is impressive.
3. The proposed method is reasonable and offers valuable insights to the field of AI4S. In many scientific applications, strict physical laws must be upheld. When training data is limited, models often struggle to learn these constraints in a data-driven way; thus, explicitly enforcing the constraint is a reasonable and effective solution.

**Weaknesses:**

1. While the main focus of the paper is on inference performance under a small number of sampling steps, I believe it would be beneficial to also include results under larger-step settings. This would provide a more comprehensive understanding of the method's performance when the computational budget is less constrained, and offer a clearer comparison with existing approaches in such scenarios.
2. It would be beneficial to discuss the relationship between proposed method with reinforcement learning (RL)-based fine-tuning approaches for diffusion models (e.g., DPOK, DRaFT). Compared to inference time guidance methods, they also involve fine-tuning to optimizing the reward. I’m curious about how the fine-tuning cost of RL compares to that of your iterative optimization algorithm.
3. How is the scalability of your method? Since your method involves enforcing constraints on all substructures, as the complexity or size of the system grows, does the method remain computationally feasible during training and inference?
4. Alphafold3 is an important baseline in this domain and is suggested to be included in experiment section for a more complete comparison. Additionally, the evaluation could benefit from incorporating more comprehensive metrics such as TM-score and pb-valid.
5. To facilitate reproducibility and enable further research based on this work, I strongly encourage the authors to release the code (such as in an anonymous form during the review period).

**Questions:**

1. In Equation (2), Is there a redundant factor of 0.5 in the definition of E(x)? It seems to be inconsistent with Equation (3).
2. The iterative linearization of eq.4 (Theorem D.1) relies on multiple approximations (replacing K by I; assume g = 0). Could you please discuss how these approximations affect the convergence speed and solution quality of the optimization problem? Maybe by numerical simulations or tests on real protein datasets) could help validate the affect of these approximations.
3. Could there be additional textual description to clarify the "invalid" structures in Figure 4? It is not always obvious which structural issues are presented: Are all cases attributed to atomic clashes, or are there other problems (e.g., bond length violations, steric hindrance)?
4. How does the method perform without relying on the Protenix-mini sampling strategy and compare it to Boltz-1?
5. While the paper emphasizes reducing the number of sampling steps from 5 to 2, it is unclear why this results in a wall-clock time reduction significantly greater than 2.5× in Figure 5 (right) when compared to Protenix-mini.

---

> ### Author Response · Authors · 2025-11-20
> **Rebuttal (1/2)**
>
> **W1: Evaluation under larger-step settings.**
>
> Our projection module readily supports varying diffusion schedules. To demonstrate this, we present a comparison with Boltz-1 and Boltz-2 across several step budgets (2, 5, 10, and 200) on the CASP15 benchmark. As shown in Fig. 7 (Appendix E), our method consistently improves upon the Boltz-1 baseline, delivering higher structural accuracy and 100% physical validity at every step count.
>
> While Boltz-2 performs the best overall, this gap stems from the underlying model architecture rather than the projection method itself. Our model relies on Boltz-1 weights and training data, whereas Boltz-2 leverages architecture changes (e.g., the affinity prediction module) and a far more extensive training dataset. These advantages are clearly reflected in its dominant interface metrics (iLDDT) across benchmarks as shown in the other experiments in the paper.
>
> Crucially, our Gauss-Seidel projection scheme is model-agnostic and compatible with any AlphaFold3-style architecture. Although full fine-tuning on Boltz-2 was outside the scope of this rebuttal, we validated this compatibility by integrating our module as a post-processing for Boltz-2. The results are shown in Fig. 7 (Appendix E). It confirms that this hybrid approach eliminates physical violations, even those seen in Boltz-2 at low step budgets (<10), while preserving the base model's high accuracy.
>
> ---
>
> **W2: Relationship with RL-based Finetuning Approaches.**
>
> We thank the reviewer for pointing this out. We have added a new Related Work subsection in the updated manuscript discussing these methods. Specifically, while both our method and RL approaches involve fine-tuning the base diffusion model to optimize a reward/loss function, the mechanisms differ significantly: our approach directly optimizes the atom coordinates through a Gauss-Seidel projection module enforcing physical constraints. This is a form of **direct** optimization for physical validity. For RL-based methods, they typically optimize a long-term reward signal (e.g., high predicted affinity or docking score) derived from the final output, often using techniques like Policy Optimization. This **indirect** optimization can be more computationally demanding and hard to converge during training compared to our direct projection approach.
>
> ---
>
> **W3: Scalability and computational cost with increasing system size.**
>
> To rigorously address this, we conducted a detailed computational cost analysis of the projection module during training, benchmarked using the CASP15 dataset. We established input complexity by binning the samples based on the number of tokens, and we reported the wall-clock runtime and memory consumption (both forward and backward passes) on an A100 GPU.
>
> The results, presented in Fig. 6 (Appendix D), validate the high efficiency and scalability of our module: i) Both the forward and backward projection passes are completed within a matter of seconds, demonstrating low latency even for the most complex inputs (maximal token bins). ii) The additional memory required for the module is below 3.5 GB across all tested bins.
>
> This high computational efficiency is crucial and is achieved through our specialized, highly optimized GPU implementation, which uses custom CUDA kernels for the projection module.
>
> ---
>
> **W4: AlphaFold3 comparison and more metrics**
>
> We have incorporated AlphaFold3 into our evaluation. These results have been added to Table 1. We observe that AlphaFold3 generally yields performance comparable to other state-of-the-art 200-step diffusion baselines, such as Boltz-1 and Protenix.
>
> As the reviewer suggested, we have extended our analysis to include TM-scores for all six datasets across all compared methods. These detailed results are now reported in Table 3 (Appendix B). Notably, our approach achieves state-of-the-art TM-scores among few-step methods.
>
> Regarding Physical Validity, our evaluation prioritizes the metric defined in the original Boltz-1 paper. We chose this because Boltz-1 serves as our base model and defines the specific physical constraints our projection module is implemented to enforce. Crucially, the PoseBusters-validity metric (pb-valid) contains a constraint set that does not perfectly align with the Boltz-1 definitions. Therefore, directly comparing PoseBusters‑valid success rates across Boltz‑1 and its fine-tuned variants (including ours) does not constitute a fair assessment of the physically valid system for which our method is optimized. Incorporating all the constraints proposed in both Boltz‑1‑steering and PoseBusters‑valid metric will be left as promising future work.

---

> > ### Author Response · Authors · 2025-11-20
> > **Rebuttal (2/2)**
> >
> > **W5: Open-sourcing Code.**
> >
> > To facilitate reproducibility, as requested by the reviewer, we have attached an anonymous GitHub repository containing the drop-in PyTorch implementation of both forward and backward processes of our Gauss-Seidel projection module (https://github.com/anonymousid14624iclr2026/id14624_rebuttal_iclr2026/). We will also open source all the training code along with all experimental results for reproducibility purposes upon acceptance.
> >
> > ---
> >
> > **Q1: Factor of 0.5 in Eq. 2.**
> >
> > The factor of 0.5 in Equation 2 is necessary because this term represents a quadratic penalty energy. When the first-order optimality condition (i.e., taking the derivative) is applied to this term, the exponent rule of differentiation will introduce a factor of $2$. The initial factor of $0.5$ cancels this $2$.
> >
> > ---
> >
> > **Q2: Convergence speed and solution quality of the iterative linearization of Eq. 4.**
> >
> > To empirically validate how replacing the matrix $K$ with the identity $I$ and assuming a negligible gradient $g \approx 0$ affects the solver, we conducted a numerical comparison on the CASP15 dataset. We implemented a full Newton baseline that uses the Conjugate Gradient method without any approximations on $K$ or $g$ and compared it against our Gauss-Seidel method. We computed the pairwise RMSD between the final structures optimized by our method and the exact full Newton baseline. The average pairwise RMSD is 0.0003 $\pm$ 0.0001 Å. This negligible difference confirms that the approximations do not alter the final converged solution much. Regarding the convergence speed, the full Newton method converges in fewer iterations (typically 5–10 steps) compared to our method (typically 10–30 steps). However, calculating and inverting the full matrix in the Newton method scales with system size ($O(N^3)$), making it prohibitively slow for large protein complexes. In contrast, our method, which relies on these approximations, avoids dense matrix operations ($O(N)$), resulting in a $\sim1.5\times$ speedup in total projection time.
> >
> > ---
> >
> > **Q3: Additional textual description for Figure 4.**
> >
> > We have updated the caption and text accompanying Fig. 4 to explicitly identify the specific physical violations shown: The first example in Fig. 4 demonstrates a ligand-protein steric clash, where the ligand atoms physically penetrate the receptor surface. This corresponds to constraint 1 (Steric Clash) in Fig. 8. The second example demonstrates an inter-chain overlap, where atoms from Chain A and Chain B occupy the same space. This corresponds to constraint 6 (Overlapping Chains) in Fig. 8.
> >
> > While our method addresses various constraints (including bond lengths and angles), the visual examples in Fig. 4 primarily highlight steric clashes (both ligand-protein and protein-protein), as these are the most visually prominent and severe failures observed in baseline models. We have revised the figure caption to provide these specific descriptions.
> >
> > ---
> >
> > **Q4: Performance without using the Protenix-mini sampling strategy**
> >
> > We acknowledge that the Protenix‑Mini ODE sampler is indeed a critical component for high-quality results in the few-step regime. To verify this, we evaluated both Boltz-1 and our method without the specific ODE sampling strategy (using standard sampling instead).
> >
> > In both cases, we observed a sharp decline in performance (Complex LDDT on CASP15): Boltz-1 dropped from 0.62 to 0.02, and ours dropped from 0.62 to 0.03. These results indicate that the ODE sampler is essential for maintaining structural accuracy when the step count is reduced. Consequently, we treat the sampler as a necessary foundation for few-step generation, upon which our Gauss-Seidel projection module operates to further ensure physical validity and refine accuracy.
> >
> > ---
> >
> > **Q5: Clarification on wall-clock runtime speedup compared to Protenix-mini.**
> >
> > We thank the reviewer for identifying this discrepancy. We confirm that this was a labeling typo in Fig. 5: the reported speedup was calculated against the standard Protenix baseline (200 steps), not Protenix‑Mini. For completeness, we also quantified the speedup relative to the actual Protenix‑Mini (5 steps). In this comparison, our method achieves a $\approx$ 2.3× speedup. We have corrected the label in Fig. 5 in the revised manuscript.

---

> ### Author Response · Authors · 2025-11-25
>
> Hi Reviewer iHkC, many thanks for your time and constructive feedback. As the discussion period is closing soon, we wanted to make sure we haven’t missed any remaining concerns, especially regarding larger-step settings, positioning vs RL-based fine-tuning approaches, scalability, and the possibility of adding stronger baselines/metrics, as well as the specific technical questions you raised (equation factor, approximation impact, and runtime explanation). If there are new questions or anything you’d like us to clarify, we’re very happy to follow up. If our rebuttal has addressed the main points, we’d be grateful if you consider updating your evaluation to match your current assessment.

---

### Official Review · Reviewer_3NuJ · 2025-10-30

**Soundness:** 2
**Presentation:** 3
**Contribution:** 2
**Rating:** 4
**Confidence:** 5

**Summary:**

This paper tackles the critical issue of physical invalidity (steric clashes, distorted geometry) in structures generated by deep learning models for biomolecular interactions, particularly all-atom diffusion models. The authors propose a method to enforce physical validity as a hard constraint during both training and inference. The core idea is a differentiable projection module that maps the provisional atomic coordinates generated by a diffusion model onto the nearest physically valid configuration . This projection is efficiently implemented using a Gauss-Seidel iterative scheme, exploiting the locality of physical constraints . Crucially, the module is made differentiable via implicit differentiation, allowing it to be integrated seamlessly into existing diffusion frameworks (like Boltz-1) for end-to-end finetuning . A key result is that incorporating this module enables the generation of physically valid and structurally accurate complexes using only 2 denoising steps, achieving accuracy comparable to 200-step SOTA baselines while offering ~10x speedup and guaranteeing validity . The method is evaluated on six diverse benchmarks against strong baselines.

**Strengths:**

- Generating physically plausible structures is a prerequisite for the reliability and utility of biomolecular models. This paper directly confronts the common failing of deep generative models in this regard , offering a principled solution.
- The use of the Gauss-Seidel method for the projection step is well-suited for the problem, leveraging the local nature of physical constraints (bond lengths, angles, clashes) for fast and stable convergence compared to methods like gradient descent.
- Making the iterative Gauss-Seidel solver differentiable via implicit differentiation  is a key technical contribution, enabling end-to-end training and allowing the diffusion model to adapt to the projection step. This integration is crucial for maintaining high accuracy, as shown in the ablation study.

**Weaknesses:**

- The evaluation of the Protenix baseline appears to be an underestimation. According to the Protenix technical report, as well as anecdotal user feedback, its performance at 200 steps (e.g., in terms of DockQ and validity metrics) is reportedly not as low as presented in this paper. I recommand the authors to either provide the detailed configuration (config) files used for their Protenix experiments or, preferably, release the raw prediction files generated by their baseline models to ensure a fair and reproducible comparison.
- Regarding the Protenix mini tech report, the original authors claim that 'increasing the ODE sampling steps to 10 effectively mitigates this issue (clash)'. Given this, the comparison in Table 1 might be suboptimal. I strongly suggest that the authors re-evaluate the baseline using 10 ODE steps in Table 1, as this seems to be the recommended setting for mitigating structural clashes.
- The paper states that GS projection achieves good physical constraint effects within just 2 steps. This raises a question: why did the authors not experiment with applying the projection for more steps? Furthermore, the results in Table 1 indicate that sampling for 2 steps with constraints still underperforms the original 'boltz2' baseline. The comparison is incomplete. The authors should also include experiments comparing their method against baselines like 'boltz2' when restricted to a similar small step budget (e.g., 10 steps) for structure prediction.
- Although the 2-step process is much faster overall, the paper could provide more details on the computational cost (time and memory, forward and backward) of the Gauss-Seidel projection module itself, especially as the system (complex) size grows larger. The backward pass involves solving a linear system using CG, which could become expensive.

**Questions:**

- Can the authors provide metrics (e.g. RMSD) quantifying the magnitude of coordinate changes introduced by the projection step? Are there cases where projection significantly alters key interface interactions or secondary structure elements?

---

> ### Author Response · Authors · 2025-11-20
> **Rebuttal (1/2)**
>
> **W1: Clarification on the underestimated Protenix baseline evaluation.**
>
> We appreciate the reviewer for raising this important point regarding the Protenix baseline. We have carefully re-investigated our evaluation pipeline to ensure fairness. It is important to first clarify that our metric implementations strictly adhere to those used in the original Boltz-1 paper.
>
> The perceived discrepancy in performance arises primarily from three factors: (1) subtle differences in metric definitions (e.g., specific L-RMSD implementations); (2) a lack of overlap between the reporting protocols (both datasets and metrics being used) of Boltz-1 and Protenix; and (3) differences in output formats. We provide a detailed clarification below:
>
> (1) There are three distinct sets of metrics used across the paper: i) LDDT variants (Complex LDDT, Prot‑Prot iLDDT, Lig‑Prot iLDDT, DNA‑Prot iLDDT, RNA‑Prot iLDDT), ii) docking-related metrics (DockQ>0.23, Mean LDDT‑PLI, L‑RMSD < 2Å), and iii) Physical validity. We provide the analysis one by one:
>
>    i) For LDDT, our evaluation aligns with Protenix. This is evidenced by the iLDDT performances on the dsDNA and RNA‑Protein datasets. Our results match those reported in Fig. 1 C/D of the original Protenix paper, confirming that our LDDT calculation is consistent.
>
>    ii) For docking-related metrics, we clarify that we calculated the numbers using OpenStructure, which strictly follows the original Boltz‑1 paper. In contrast, the Protenix technical report uses PXMeter. This causes absolute value differences between our reported numbers and those in the Protenix paper. We re-evaluated the [L‑RMSD < 2Å] metric using PXMeter on PoseBusters for all methods (see Table 4 in Appendix G) and found the exact number of the Protenix baseline aligns closely with the Protenix original paper. Crucially, while the **absolute** values shift between PXMeter and OpenStructure depending on the tool used, the **relative** ranking of the methods **remains** unchanged. Thus, we argue that the docking-related metrics in our manuscript remain valid for comparison.
>
>    iii) Regarding the PoseBusters-valid success rate, our reported Physical Validity metric strictly adheres to the original Boltz‑1‑steering paper, which serves as the base model for our fine-tuning. We prioritized this metric to ensure the evaluation aligns with the specific constraints our projection module is designed to enforce. Crucially, the PoseBusters-valid metric includes constraints that do not fully overlap with those in Boltz‑1‑steering, and our current projection module does not implement these additional constraints. Consequently, a direct comparison of PoseBusters-valid success rates across Boltz‑1 and its fine-tuned variants (including ours) would not fairly assess the specific optimization goals of our method. We view the integration of all constraints from both Boltz‑1‑steering and PoseBusters as a promising direction for future work.
>
> (2) There is limited overlap between the reporting protocols of Boltz-1 and Protenix: Protenix originally reported LDDT only on the RNA targets of CASP15, whereas Boltz‑1 reports on the full dataset; Boltz‑1 reports exclusively on CASP15 and Test with all the metrics, whereas Protenix reports partial metrics on different datasets ([L‑RMSD < 2Å] on the PoseBusters dataset, DNA‑Prot iLDDT on the dsDNA dataset, RNA‑Prot iLDDT on the RNA‑Protein dataset). Our paper evaluates **all** datasets with **all** metrics consistently across **all** methods to bridge this gap.
>
> (3) For the differences on the output formats, it only occurs on CASP15. We acknowledge the lower Protenix performance on CASP15 initially observed by the reviewer. Upon investigation, we found a mismatch in inference scope: We evaluated Protenix by predicting **all** chains of each target, whereas Boltz evaluates only a specific **subset** of chains defined in CASP15. This mismatch caused inconsistent comparisons. We have aligned the Protenix output with Boltz during the rebuttal and updated Table 1. With this correction, Protenix achieves performance comparable to other 200-step methods.
>
> Additionally, consistent with Protenix-Mini, we only evaluated protein complexes with a token count under 1176 due to GPU memory limitations; however, we ensured the set of evaluated protein complex IDs is consistent across all baselines.
>
> To facilitate reproducibility, as requested by the reviewer, we have attached an anonymous GitHub repository containing the specific configuration files used for our Protenix experiments (https://github.com/anonymousid14624iclr2026/id14624_rebuttal_iclr2026/). We commit to releasing the full evaluation results and raw prediction files upon acceptance, ensuring full transparency while adhering to ICLR double-blind review requirements.

---

> > ### Author Response · Authors · 2025-11-20
> > **Rebuttal (2/2)**
> >
> > **W2: Re-evaluation of Protenix-mini baseline with 10 ODE Steps.**
> >
> > We added Protenix‑Mini with 10 ODE steps across all six datasets and updated Table 1 accordingly. As suggested, 10‑step Protenix‑Mini improves both Physical Validity and structural accuracy relative to the 5‑step setting. However, it still does not guarantee physical validity, and our method remains notably stronger on CASP15, Test, PoseBusters, and AF3‑AB in LDDT‑related metrics, while using only two denoising steps with guaranteed physical validity.
> >
> > ---
> >
> > **W3: Gauss-Seidel projection with more diffusion steps and comparisons with baselines using smaller step budget.**
> >
> > Scaling our projection module to support varying diffusion steps is straightforward. To provide a comprehensive analysis, we conducted experiments comparing Boltz-1, Boltz-2, and our method (built on Boltz-1) across different diffusion step budgets (2, 5, 10, and 200) on the CASP15 dataset. As shown in Fig. 7 (Appendix E), our method demonstrates consistent improvement over the Boltz-1 baseline, outperforming it in terms of both structural accuracy and physical validity across all step budgets.
> >
> > We acknowledge that our current implementation does not surpass the performance of Boltz-2. We attribute this to the underlying base model: our method is fine-tuned using the Boltz-1 architecture, checkpoint weights, and dataset. In contrast, Boltz-2 benefits from significant architectural advancements (e.g., the affinity prediction module) and a much larger training dataset, particularly focused on protein complex interactions. This advantage is evident in the interface metrics (iLDDT) reported throughout our paper (especially Table 2), where Boltz-2 outperforms all methods by a large margin.
> >
> > Theoretically, our Gauss-Seidel projection module is agnostic to the base model and can be integrated into any AlphaFold3-like architecture, including Boltz-2. While full fine-tuning with Boltz-2 was not feasible within the rebuttal window, we validated this potential by applying our projection module as a post-processing on top of Boltz-2. Results in Fig. 7 (Appendix E) show that this hybrid approach effectively ensures 100% physical validity, resolving invalid outputs observed in Boltz-2 at low step budgets (<10), while maintaining the high structural accuracy of the original model. We thus treat our method as a general-purpose physical validity enforcement module that can be flexibly integrated into various diffusion-based structure prediction frameworks.
> >
> > ---
> >
> > **W4: Details on the computational cost of the Gauss-Seidel projection module itself.**
> >
> > We have conducted a detailed analysis of the computational cost of the Gauss-Seidel projection module during training, benchmarked on the CASP15 dataset. To provide a canonical measure of input complexity, we binned the samples by the number of tokens and reported the forward and backward wall-clock runtime and memory usage on an A100 GPU.
> >
> > The comprehensive results, detailed in Fig. 6 (Appendix D), demonstrate the module's high efficiency. Both the forward and backward passes are completed within a matter of a few seconds, and the additional memory overhead is held below 3.5 GB, even for the maximal token bins. This high efficiency is achieved by our specialized GPU implementation, leveraging custom CUDA kernels for both the forward and backward processes.
> >
> > ---
> >
> > **Q1: Metrics quantifying the magnitude of coordinate changes introduced by the projection step with case studies.**
> >
> > We have computed the RMSD before and after the projection module on the PoseBusters dataset to quantify the magnitude of coordinate changes. The average RMSD across the dataset is 0.1865 $\pm$ 0.0219 Å, confirming that the projection step introduces atomic position changes sufficient to enforce physical validity.
> >
> > To illustrate the effect of these changes, we provide four case studies, shown in Fig. 8 (Appendix H), alongside the RMSD value for each specific case. For PDB 7P4C, the projection restored a distorted $\alpha$-helix that was originally clashing with a neighboring strand. In PDB 7PL1, a self-entanglement within a coil region was successfully disentangled. Similarly, for PDB 7ELT and PDB 8AEM, the module resolved ligand‑sidechain and intra‑ligand steric clashes. These visualizations demonstrate that the projection module effectively resolves clashes between secondary structure elements and improves the physical validity of interfaces.

---

> ### Author Response · Authors · 2025-11-25
>
> Hi Reviewer 3NuJ, many thanks for your time and detailed comments. As the discussion period is closing soon, we wanted to ensure we haven’t missed any remaining concerns, particularly on the Protenix(-mini) baseline configuration/fairness (including step settings), adding small-step budget baselines (e.g., boltz2 at comparable steps), and additional details on projection impact/cost (including the magnitude of coordinate changes). If there are new questions or anything you’d like us to clarify, we’re very happy to follow up. If our rebuttal has addressed the main points, we’d be grateful if you consider updating your evaluation to match your current assessment.

---

> > ### Comment · Reviewer_3NuJ · 2025-11-26
> >
> > Thanks for the authors' rebuttal and Zhenyu's comment. They address most of my concerns. I've raised my score.

---

### Official Review · Reviewer_66sZ · 2025-11-10

**Soundness:** 3
**Presentation:** 3
**Contribution:** 3
**Rating:** 6
**Confidence:** 4

**Summary:**

This paper introduces a physically constrained diffusion framework for biomolecular structure generation that enforces local geometric and energetic constraints while maintaining structural accuracy. The method integrates a Gauss–Seidel based differentiable projection that iteratively enforces constraints to ensure the physical feasibility of generated structures. The framework supports backpropagation through conjugate gradients, enabling stable gradient-based training and inference. Empirical results demonstrate improved geometric validity, structural stability, and fast inference across complex biomolecular benchmarks.

**Strengths:**

- The motivation is clear and well grounded in the need for physically valid biomolecular generation.
- The method enforces atomically realistic outputs compared to unconstrained baselines, a practically important contribution.
- The algorithmic formulation is sound. The iterative Gauss–Seidel projection and penalty method are mathematically principled and differentiable.

**Weaknesses:**

- The PoseBusters results show a notable drop in docking-related metrics, but the paper does not analyze the cause, possibly due to a trade-off between hard constraint enforcement and ligand flexibility.
- Some evaluation metrics are missing. Including measures such as the PoseBusters-valid success rate, TM-Score, and iLDDT (in Table 2) would make the empirical validation more complete and convincing.

**Questions:**

1. Since the Gauss–Seidel scheme guarantees uniqueness only for each linearized subproblem, can different orderings or initialization seeds lead to non-unique final projections? Have the authors observed multiple feasible solutions in practice?
2. The drop in docking-related metrics on PoseBusters is notable. Can the authors provide analysis or ablation evidence on whether this degradation stems from the specific constraint choices or the limited number of denoising steps?
3. Why is metrics such as iLDDT excluded from Table 2? Including complementary metrics such as TM-Score or PoseBusters-valid success rate would make the evaluation more comprehensive.
4. Could the authors clarify how $\alpha$ is chosen in practice, and whether convergence depends sensitively on this value?

---

> ### Author Response · Authors · 2025-11-20
> **Rebuttal (1/2)**
>
> **W1 & Q2: Performance drop on PoseBusters in docking-related metrics.**
>
> Thank you for pointing this out. We investigated the performance drop on PoseBusters and found it is indeed driven by the constraint weight $\alpha$, which controls how tightly the projection enforces constraints relative to ligand pose adjustments. A too‑large weight can over‑restrict ligand rearrangements in the pocket, slightly hurting docking‑related metrics.
>
> During rebuttal, we tuned the parameter $\alpha$ on PoseBusters and observed that a slightly smaller $\alpha({=}10^{-6})$ alleviates this over‑restriction while keeping physical validity. This adjustment improves the docking metric [DockQ>0.23] from 0.53 ± 0.05 to 0.58 ± 0.10, making our performance comparable to other baselines such as Boltz‑1‑Steering (0.59 ± 0.10) and Protenix‑Mini (0.60 ± 0.10). We have adopted this $\alpha{=}10^{-6}$ for PoseBusters and updated Table 1 accordingly. A more detailed analysis of $\alpha$ is included in the answer to **Q4**.
>
> ---
>
> **W2 & Q3: Additional evaluation metrics (PoseBusters‑valid success rate, TM‑Score, and iLDDT).**
>
> We have updated Table 2 to report iLDDT scores across the three interface datasets and clarified the specific definition used per task: AF3‑AB (protein–protein/antibody iLDDT), dsDNA–Protein (DNA–protein iLDDT), and RNA–Protein (RNA–protein iLDDT). Our method achieves the best iLDDT among all few‑step methods on the AF3‑AB and dsDNA datasets, and notably outperforms the 200‑step methods (Boltz‑1 and Boltz‑1‑Steering) on the RNA–Protein dataset.
>
> For TM‑Score, we now include a complete comparison for all datasets in Table 3, Appendix B. Our approach is state‑of‑the‑art in the few‑step regime and comparable to 200‑step methods.
>
> Regarding the PoseBusters‑valid success rate, our reported **Physical Validity** metric is defined to strictly follow the original Boltz‑1‑steering paper, which provides the base model and training dataset for our fine-tuning. We prioritized this metric to ensure the evaluation remains aligned with the specific constraints our projection module is designed to enforce. Importantly, the constraints comprising the PoseBusters‑valid metric do not fully overlap with those used in Boltz‑1‑steering, and we did not implement PoseBusters‑specific constraints within our projection module. Consequently, directly comparing PoseBusters‑valid success rates across Boltz‑1 and its fine-tuned variants (including ours) does not constitute a fair assessment of the physically valid system for which our method is optimized. Incorporating all the constraints proposed in both Boltz‑1‑steering and PoseBusters‑valid metric will be left as promising future work.
>
> We also note that Boltz‑2 outperforms all other methods regarding interface-related metrics. We attribute this to its pretraining on a much larger, binding‑centric dataset, which provides a stronger prior for inter‑biomolecular interactions. Our own experimental results are derived from fine-tuning the Boltz‑1 weights and its training dataset. Crucially, even when starting from this weaker base, our method consistently surpasses the original Boltz‑1 across all datasets. Furthermore, our constraint projection module is orthogonal to the base model's architecture and pre‑trained weights; it is therefore complementary to the base model’s data priors. We expect that applying our module when fine‑tuning from Boltz‑2 would further lift docking and binder metrics, which we leave as a promising direction for future work. We have also added a section discussing this in Appendix E in the updated manuscript.

---

> > ### Author Response · Authors · 2025-11-20
> > **Rebuttal (2/2)**
> >
> > **Q1: Sensitivity of Gauss–Seidel scheme to different orderings and initialization seeds**
> >
> > Theoretically, while ordering and initialization govern the convergence speed of the Gauss–Seidel scheme, they do not influence the solution's uniqueness [1]. This holds true because the linear system solved in our method is strictly positive definite (see Appendix J).
> >
> > We investigated this empirically on the PoseBusters dataset by performing five independent projection runs for each denoised structure. The computed pairwise RMSD among outputs was 0.0000 $\pm$ 0.0000 Å. This confirms that, in practice, the projection module behaves as a stable mapping, reliably converging to a numerically identical solution. We have added this discussion in Appendix C.
> >
> > [1] Golub, Gene H.; Van Loan, Charles F. (1996), *Matrix Computations*
> >
> > ---
> >
> > **Q4: Practical choice of $\alpha$ and whether convergence is sensitive to this value?**
> >
> > We conducted an ablation study of $\alpha$ on protein PDB 7WUY and evaluated the convergence behavior. The results are added in Appendix D. Empirically, an excessively large $\alpha$ fails to converge to a physically valid configuration because the constraint penalties are not enforced strongly enough. Conversely, when alpha is smaller than $10^{-5}$, the projection consistently converges to a valid state.

---

> ### Author Response · Authors · 2025-11-25
>
> Hi Reviewer 66sZ, many thanks for your time and thoughtful feedback. As the discussion period is closing soon, we wanted to check whether any concerns remain after our rebuttal (e.g., potential non-uniqueness under different GS orderings/initializations, the PoseBusters docking-metric drop, and the additional metrics you suggested, as well as the practical choice of the relevant hyperparameter). If there are new questions or anything you’d like us to clarify, we’re very happy to follow up. If our rebuttal has addressed the main points, we’d be grateful if you consider updating your evaluation to match your current assessment.

---

### Official Review · Reviewer_YVMA · 2025-11-19

**Soundness:** 3
**Presentation:** 2
**Contribution:** 2
**Rating:** 6
**Confidence:** 3

**Summary:**

This paper presents a Gauss–Seidel–based projection module that enables stable and rapid convergence to physically plausible structures using only a few denoising steps. By explicitly enforcing geometric constraints during sampling, the method significantly accelerates inference while maintaining a high degree of physical validity.

**Strengths:**

1. The use of a Gauss–Seidel projection mechanism to enforce physical validity is novel and appears highly effective. It provides a computationally efficient way to ensure that generated structures adhere to key geometric constraints.

2. Both quantitative and qualitative results demonstrate that the proposed method is effective and efficient across various evaluation metrics.

**Weaknesses:**

1. The paper does not release its code, project page, or any runnable repository. This raises concerns regarding reproducibility and the ability of other researchers to validate or extend the results.

2. The experiments are conducted by finetuning the method on an existing co-folding model (Boltz-1). However, the reported two-step sampling results do not consistently surpass the original Boltz-1 baseline. Although this may be attributable to the reduced number of denoising steps, it remains unclear whether the proposed method can outperform Boltz-1 when using a comparable number of steps.

3. The comparison with Proteinix-mini appears somewhat unfair, as Proteinix-mini uses a significantly reduced architecture and removes the MSA module, placing it at an inherent disadvantage.

4. Since the second-stage projection refines the output from the first denoising step, the method improves physical validity but may have limited capacity to further enhance structural accuracy. It is unclear whether the approach can simultaneously improve both accuracy and validity, or whether it inherently trades one for the other.

**Questions:**

See weakness

---

> ### Author Response · Authors · 2025-11-20
> **Rebuttal**
>
> **W1: Code release and reproducibility.**
>
> We thank the reviewer for emphasizing the importance of reproducibility. To address this, we have attached an anonymous GitHub repository (https://github.com/anonymousid14624iclr2026/id14624_rebuttal_iclr2026/ ) containing the drop-in PyTorch implementation of both the forward and backward processes of our Gauss-Seidel projection module. Upon acceptance, we commit to open-sourcing the full training code, configuration files, and experimental results to ensure the community can validate and extend our work.
>
> **W2: Performance comparison with Boltz-1 using a comparable number of steps.**
>
> During the rebuttal, we conducted additional experiments comparing Boltz-1, Boltz-2, and our method across different step budgets (2, 5, 10, and 200) on the CASP15 dataset. As shown in Fig. 7 (Appendix E), our method consistently outperforms the original Boltz-1 baseline in terms of both structural accuracy (Complex-LDDT) and physical validity across all tested step budgets, including the 200-step setting. This confirms that our method provides improvements regardless of the sampling budget, rather than trading accuracy for speed.
>
> **W3: Fairness of comparison with Protenix-Mini.**
>
> We acknowledge that Protenix-Mini uses a reduced architecture, which is a lighter model than the Boltz-1 backbone we use. However, we believe the comparison is fair and valuable for the following reasons: 1) Our goal is to demonstrate that by using our projection module, a heavy, high-accuracy model (Boltz-1) can be accelerated to run at an even faster speed (2 steps) as a lightweight model (Protenix-Mini) while retaining the superior structural accuracy of the heavy model. 2) To ensure a balanced evaluation against high-capacity models, we also compare against the full Protenix model (200 steps) in Table 1. Our 2-step method achieves competitive or superior structural accuracy to these computation-heavy baselines while delivering a $\sim10\times$ wall-clock speedup.
>
> **W4: Trade-off between physical validity and structural accuracy.**
>
> We appreciate the reviewer’s concern regarding this potential trade-off; however, our empirical data suggests that the two objectives are not mutually exclusive in our framework. We provide both qualitative and quantitative evidence that our method effectively decouples these responsibilities to improve both:
>
> 1. As illustrated in Fig. 3 of our manuscript, simply applying the projection without fine-tuning can sometimes disrupt secondary structures (e.g., the broken $\alpha$-helix in the left figure) in the effort to satisfy constraints. However, our key contribution is making the Gauss-Seidel projection differentiable and fine-tuning the diffusion model through it. This allows the diffusion network to learn to focus exclusively on recovering the global structural accuracy, effectively "offloading" the task of ensuring physical validity to the projection layer. As shown in Fig. 3 Right, the fine-tuned model successfully recovers the correct $\alpha$-helical structure that was otherwise lost, demonstrating that the projection layer acts as a synergistic guide rather than a limitation. The preservation of high structural accuracy and physical validity is also evidenced by quantitative results in Tables 1 and 2.
>
> 2. As detailed in the response to W2, when comparing at equal step budgets (e.g., 200 steps), our method achieves higher Complex-LDDT scores than the baseline Boltz-1. By resolving physical invalid states and using the "decoupling" strategy described above, we enhance structural quality alongside physical validity.

---

> ### Author Response · Authors · 2025-11-25
>
> Hi Reviewer YVMA, many thanks for your time and detailed comments. As the discussion period is closing soon, we wanted to make sure we haven’t missed any remaining concerns (especially around reproducibility/code release, the step-budget comparison to Boltz-1, and the fairness of baselines like Protenix-mini). If there are new questions or anything you’d like us to clarify, we’re very happy to follow up. If our rebuttal has addressed the main points, we’d be grateful if you consider updating your evaluation to match your current assessment.

---

### Author Response · Authors · 2025-11-20
**General Response**

We sincerely appreciate the detailed reviews and the thoughtful feedback provided by all reviewers. In addition to addressing specific comments from each reviewer, we would like to outline our primary contributions.

+ **[Motivation]** We address **the critical challenge of physical invalidity** inherent in biomolecular structures generated by current all-atom diffusion models [66sZ, 3NuJ, iHkC].

+ **[Methodology]** Our approach is recognized as **novel and theoretically sound**, offering a **principled** solution that rigorously enforces physical constraints within the generative process [YVMA, 66sZ, 3NuJ, iHkC].

+ **[Experiments]** Extensive empirical results demonstrate both **superior effectiveness and efficiency**, highlighting significant inference acceleration while maintaining high structural accuracy [YVMA, 66sZ, 3NuJ, iHkC].

During the rebuttal period, we have made the following revisions to the manuscript as recommended by the reviewers, as highlighted in red in the uploaded PDF:

* A comprehensive ablation study comparing Boltz-1, Boltz-2, and our method across varying diffusion steps (2, 5, 10, and 200) (Appendix E and Fig. 7).

* A detailed computational cost analysis (runtime and memory overhead) of the Gauss-Seidel projection module benchmarked on CASP15 (Appendix F and Fig. 6).

* An ablation study on the practical choice of $\alpha$ and its sensitivity of convergence (Appendix D and Fig. 6).

* Extended evaluation metrics, including TM-Scores for all six datasets and iLDDT scores for interface datasets (Table 3 and 4, Appendix G).

* Added 10-step Protenix-Mini and AlphaFold3 as baselines (Tables 1 and 2).

* Aligned CASP15 evaluation for Protenix and docking-related metrics (Table 1).

* A quantitative analysis of structural corrections along with qualitative visualizations of the Gauss-Seidel projection (Appendix H and Fig. 8).

* **Released an anonymous GitHub repo** with a drop‑in PyTorch implementation (forward and backward) of the Gauss–Seidel projection module; included configuration files for our Protenix experiments: https://github.com/anonymousid14624iclr2026/id14624_rebuttal_iclr2026/

* Incorporated all recommended references, discussions on RL-based fine-tuning, suggestions on paper improvement, and clarifications on metric implementations and figure labels.

We hope our responses address all reviewers' concerns and help improve the review scores. We thank all reviewers and the AC again for their time and efforts!

---

### Public Comment · ~Zhenyu_Liu19 · 2025-11-20
**Question about physical validity evaluation protocol**

Dear authors,

I am a member of the Protenix team. Thank you for citing our work and for pushing forward on the important issue of physical validity in structure prediction.

I noticed in Table 1 that Protenix and other base models without steering have relatively low physical validity scores. For example, the physical validity of AlphaFold3 on the PoseBusters dataset is reported as 0.45, which is far below the numbers in their original paper and somewhat surprising to me.

To better understand your evaluation, could you clarify which implementation you used to compute the physical validity metrics in your protocol?  In particular, if you used the code from the *boltz* repository, there is a subtle point regarding the tetrahedral chirality metric that might be worth double-checking. As far as we can tell, when computing chiral atom violations, that code directly uses the chirality annotations from the CCD mol and relies on a fixed atom ordering. For groups like `CO[P@@](=O)([O-])O[P@](=O)(O)OC` (for example in CCD_NAP), Protenix and some other base models treat the `(=O)` and `(O)` atoms as permutable. If the evaluation matches atoms only by name in such cases, this can overestimate the true number of chirality violations.

Thank you for your time and for any clarification you can provide. We believe that resolving this discrepancy could be helpful both for your paper and for the broader community interested in physically valid biomolecular structure prediction.

---

> ### Author Response · Authors · 2025-11-21
>
> Hi Zhenyu,
>
> Thank you for your insightful question. First, we would like to express our gratitude to the Protenix team for their open-source contributions. Your work and the release of reproducible baselines have been instrumental to the community and our own research.
>
> Please find our detailed clarification below regarding the evaluation metrics:
>
> (1) We used the Boltz physical validity metric because our model is finetuned from the Boltz-1 checkpoint, and adhering to their evaluation protocol ensures consistency. It is important to distinguish the scope of these metrics: the Boltz-1 "physical validity" evaluates the **entire protein complex**, whereas the PoseBusters-valid score (e.g., in Fig. 4 of the AlphaFold3 paper) evaluates validity specifically for the **ligand and its immediate protein neighborhood**. This fundamental difference in scope naturally leads to numerical discrepancies between the two.
>
> (2) As also discussed in our response to Reviewer 3NuJ (W1), the Boltz physical validity metric is considerably stricter than standard PoseBusters-valid. While it incorporates the overlapping checks found in PoseBusters (e.g., bond lengths/angles, internal clashes, chirality, flatness, ligand–protein clashes), it also enforces additional constraints not covered by PoseBusters-valid, such as steric clashes between protein chains.
>
> To verify our benchmarking on AlphaFold3, we evaluated our computed ouputs from AlphaFold3 using the standard PoseBusters-valid metric directly. We obtained a score of 0.7745 ± 0.0612, which aligns closely with the values reported in the AlphaFold3 paper (Fig. 4). Similarly, our reproduced physical validity results for Boltz-1, Boltz-Steering, and Boltz-2 on CASP15 and TEST are consistent with those in their technical report.
>
> (3) We fully agree with your observation regarding chirality. The original Boltz evaluator relied on fixed CCD atom ordering, which indeed overcounted violations for permutable substituents. We validated this issue and implemented a fix for the chiral-center matching logic. The updated physical validity scores are as follows:
>
> * **AlphaFold3:** 0.4526 ± 0.0279 $\rightarrow$ 0.4969 ± 0.0675
> * **Boltz-1:** 0.2098 ± 0.0124 $\rightarrow$ 0.2889 ± 0.0667
> * **Protenix:** 0.3656 ± 0.0322 $\rightarrow$ 0.3722 ± 0.0722
>
> This confirms that the original logic inflated chirality violations. However, while these adjustments modestly shift the absolute values, they do not alter the overall qualitative conclusions or the relative ranking of the methods.
>
> We appreciate you raising this technical point and welcome any further suggestions.

---

### Meta-Review · Area_Chair_yTZc · 2026-01-07

**Summary:**

This paper targets a common failure mode of diffusion-style biomolecular structure generators. The authors propose a GS-based differentiable projection module that maps a denoised structure onto the nearest configuration satisfying a set of local physical/geometric constraints. The key engineering point is that the projection is both fast and differentiable, so the diffusion model can be fine-tuned end-to-end “through” the projection. Empirical results on many benchmarks have demonstrated the performance of the proposed research.

The concerns of the reviewers are mainly around the fairness of evaluation, selection of metrics and other technical clarity questions.

**Reviewer Concerns:**

The authors provided a response which addressed the reviewers' concern regarding some of the technical details. some of the partially opening questions including:
- the fairness of the evaluation (R. YVMA) should be evaluated by a matched capacity instead of the current adding a new baseline as comparison.
- some of the metrics (like pb-valid) is very unique to ICLR's community, the authors are suggested to evaluate their results on more public-accessible metrics.
- The relationship to the RL-based finetuning (as pointed by R. iHkC is not quantitively studied.

**Reviewer Scores:**

Reviewer 3NuJ has acknowledged that they will raise the score to 6. Thus this paper ends up with a 6/6/6/6 rating. The other reviewers are not likely to further increase their score.

---

### Decision · Program_Chairs · 2026-01-26

Accept (Poster)